# TextDiffuser: Diffusion Models as Text Painters

**Jingye Chen**[*13], **Yupan Huang**[*23], **Tengchao Lv**[3], **Lei Cui**[3], **Qifeng Chen**[1], **Furu Wei**[3]
[1]HKUST     [2]Sun Yat-sen University     [3]Microsoft Research
qwerty.chen@connect.ust.hk, huangyp28@mail2.sysu.edu.cn, cqf@ust.hk
{tengchaolv,lecu,fuwei}@microsoft.com

## Abstract

Diffusion models have gained increasing attention for their impressive generation abilities but currently struggle with rendering accurate and coherent text. To address this issue, we introduce **TextDiffuser**, focusing on generating images with visually appealing text that is coherent with backgrounds. TextDiffuser consists of two stages: first, a Transformer model generates the layout of keywords extracted from text prompts, and then diffusion models generate images conditioned on the text prompt and the generated layout. Additionally, we contribute the first large-scale text images dataset with OCR annotations, **MARIO-10M**, containing 10 million image-text pairs with text recognition, detection, and character-level segmentation annotations. We further collect the **MARIO-Eval** benchmark to serve as a comprehensive tool for evaluating text rendering quality. Through experiments and user studies, we show that TextDiffuser is flexible and controllable to create high-quality text images using text prompts alone or together with text template images, and conduct text inpainting to reconstruct incomplete images with text. The code, model, and dataset will be available at https://aka.ms/textdiffuser.

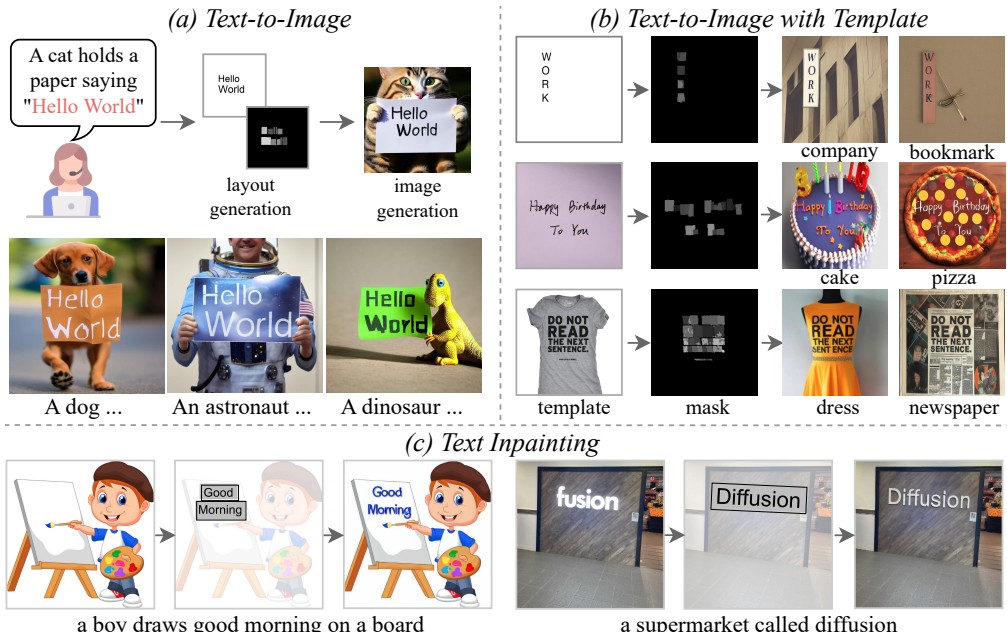

Figure 1: TextDiffuser generates accurate and coherent text images from text prompts or together with template images, as well as conducting text inpainting to reconstruct incomplete images.

---

*Equal contribution during internship at Microsoft Research.

# 1 Introduction

The field of image generation has seen tremendous progress with the advent of diffusion models [2, 15, 16, 18, 25, 67, 70, 72, 79, 93] and the availability of large-scale image-text paired datasets [17, 74, 75]. However, existing diffusion models still face challenges in generating visually pleasing text on images, and there is currently no specialized large-scale dataset for this purpose. The ability of AI models to generate accurate and coherent text on images is crucial, given the widespread use of text images in various forms (*e.g.*, posters, book covers, memes, etc.) and the difficulty in creating high-quality text images, which typically require professional skills and numerous times of designers.

Traditional solutions to creating text images involve using image processing tools like `Photoshop` to add text onto images directly. However, these often result in unnatural artifacts due to the background's complex texture or lighting variations. Recent efforts have used diffusion models to overcome the limitations of traditional methods and enhance text rendering quality. For instance, Imagen [72], eDiff-I [2], and DeepFloyd [12] observe diffusion models generate text better with T5 series text encoders [61] than the CLIP text encoder [60]. Liu et al. employ character-aware text encoders to improve text rendering [46]. Despite some success, these models only focus on text encoders, lacking control over the generation process. A concurrent work, GlyphDraw [49], improves the controllability of models by conditioning on the location and structures of Chinese characters. However, GlyphDraw does not support multiple text bounding-box generation, which is not applicable to many text images such as posters and book covers.

In this paper, we propose **TextDiffuser**, a flexible and controllable framework based on diffusion models. The framework consists of two stages. In the first stage, we use a Layout Transformer to locate the coordinates of each keyword in text prompts and obtain character-level segmentation masks. In the second stage, we fine-tune the latent diffusion model by leveraging the generated segmentation masks as conditions for the diffusion process and text prompts. We introduce a character-aware loss in the latent space to further improve the quality of generated text regions. Figure 1 illustrates the application of TextDiffuser in generating accurate and coherent text images using text prompts alone or text template images. Additionally, TextDiffuser is capable of performing text inpainting[2] to reconstruct incomplete images with text. To train our model, we use OCR tools and design filtering strategies to obtain 10 million high-quality i**ma**ge-text pai**r**s wi**t**h **O**CR annotations (dubbed as **MARIO-10M**), each with recognition, detection, and character-level segmentation annotations. Extensive experiments and user studies demonstrate the superiority of the proposed TextDiffuser over existing methods on the constructed benchmark **MARIO-Eval**. The code, model and dataset will be publicly available to promote future research.

# 2 Related Work

**Text Rendering.** Image generation has made significant progress with the advent of diffusion models [18, 25, 67, 72, 79, 63, 70, 2, 8, 13, 52, 48, 26, 80], achieving state-of-the-art results compared with previous GAN-based approaches [64, 100, 43, 58]. Despite rapid development, current methods still struggle with rendering accurate and coherent text. To mitigate this, Imagen [72], eDiff-I [2], and DeepFolyd [12] utilize a large-scale language model (large T5 [61]) to enhance the text-spelling knowledge. In [46], the authors noticed that existing text encoders are blind to token length and trained a character-aware variant to alleviate this problem. A concurrent work, GlyphDraw [49], focuses on generating high-quality images with Chinese texts with the guidance of text location and glyph images. Unlike this work, we utilize Transformer [81] to obtain the layouts of keywords, enabling the generation of texts in multiple lines. Besides, we use character-level segmentation masks as prior, which can be easily controlled (*e.g.*, by providing a template image) to meet user needs.

Several papers have put forward benchmarks containing a few cases regarding text rendering for evaluation. For example, Imagen [72] introduces DrawBench containing 200 prompts, in which 21 prompts are related to visual text rendering (*e.g.*, A storefront with 'Hello World' written on it). According to [46], the authors proposed DrawText comprising creative 175 prompts (*e.g.*, letter 'c' made from cactus, high-quality photo). GlyphDraw [49] designs 218 prompts in Chinese and

---

[2]Different from text editing [88, 37, 32], the introduced text inpainting task aims to add or modify text guided by users, ensuring that the inpainted text has a reasonable style (*i.e.*, no need to match the style of the original text during modification exactly) and is coherent with backgrounds.

English (*e.g.*, Logo for a chain of grocery stores with the name 'Grocery'). Considering that existing benchmarks only contain a limited number of cases, we attempt to collect more prompts and combine them with existing prompts to establish a larger benchmark MARIO-Eval to facilitate comprehensive comparisons for future work.

**Image Inpainting.** Image inpainting is the task of reconstructing missing areas in images naturally and coherently. Early research focused on leveraging low-level image structure and texture to address this task [3, 6, 5]. Later, deep learning architectures such as auto-encoder [55, 45], GAN [65, 98], VAE [103, 105], and auto-regressive Transformers [57, 83, 96] were applied to tackle this problem. Recently, diffusion models have been used to generate high-quality and diverse results for unconditional image inpainting [71, 48, 67, 11, 99], text-conditional image inpainting [52, 1] and image-conditional image inpainting [92]. Our work falls under the category of text-conditional image inpainting using diffusion models. In contrast to prior works that focused on completing images with natural backgrounds or objects, our method focuses on completing images with text-related rendering, also named text inpainting, by additional conditioning on a character-level segmentation mask.

**Optical Character Recognition.** Optical Character Recognition (OCR) is an important task that has been studied in academia for a long period [87, 7]. It has undergone a remarkable development in the last decade, contributing to many applications like autonomous driving [89, 73], car license plate recognition [101, 53], GPT models [76, 28], etc. Various datasets [31, 19, 90, 91] and downstream tasks are included within this field, such as text image recognition [77, 41, 95, 14], detection [106, 51, 42, 84], segmentation [90, 91, 107, 68], super-resolution [9, 85, 50, 104], as well as some generation tasks, including text image editing [88, 78, 94, 59, 38], document layout generation [56, 22, 20, 40, 33], font generation [30, 21, 36, 54], etc. Among them, the font generation task is most *relevant* to our task. Font generation aims to create high-quality, aesthetically pleasing fonts based on given character images. In contrast, our task is more challenging as it requires the generated text to be legible, visually appealing, and coherent with the background in various scenarios.

## 3  Methodology

As illustrated in Figure 2, TextDiffuser consists of two stages: Layout Generation and Image Generation. We will detail the two stages and introduce the inference process next.

### 3.1  Stage1: Layout Generation

In this stage, the objective is to utilize bounding boxes to determine the layout of keywords (enclosed with quotes specified by user prompts). Inspired by Layout Transformer [20], we utilize the Transformer architecture to obtain the layout of keywords. Formally, we denote the tokenized prompt as $\mathcal{P} = (p_0, p_1, ..., p_{L-1})$, where $L$ means the maximum length of tokens. Following LDM [67], we use CLIP [60] and two linear layers to encode the sequence as $\text{CLIP}(\mathcal{P}) \in \mathbb{R}^{L \times d}$, where $d$ is the dimension of latent space. To distinguish the keywords against others, we design a keyword embedding $\text{Key}(\mathcal{P}) \in \mathbb{R}^{L \times d}$ with two entries (*i.e.*, keywords and non-keywords). Furthermore, we encode the width of keywords with an embedding layer $\text{Width}(\mathcal{P}) \in \mathbb{R}^{L \times d}$. Together with the learnable positional embedding $\text{Pos}(\mathcal{P}) \in \mathbb{R}^{L \times d}$ introduced in [81], we construct the whole embedding as follows:

$$\text{Embedding}(\mathcal{P}) = \text{CLIP}(\mathcal{P}) + \text{Pos}(\mathcal{P}) + \text{Key}(\mathcal{P}) + \text{Width}(\mathcal{P}). \tag{1}$$

The embedding is further processed with Transformer-based $l$-layer encoder $\Phi_E$ and decoder $\Phi_D$ to get the bounding boxes $\mathbf{B} \in \mathbb{R}^{K \times 4}$ of $K$ key words autoregressively:

$$\mathbf{B} = \Phi_D(\Phi_E(\text{Embedding}(\mathcal{P}))) = (\mathbf{b}_0, \mathbf{b}_1, ..., \mathbf{b}_{K-1}). \tag{2}$$

Specifically, we use positional embedding as the query for the Transformer decoder $\Phi_D$, ensuring that the $n$-th query corresponds to the $n$-th keyword in the prompt. The model is optimized with $l1$ loss, also denoted as $|\mathbf{B}_{GT} - \mathbf{B}|$ where $\mathbf{B}_{GT}$ is the ground truth. Further, we can utilize some Python packages like `Pillow` to render the texts and meanwhile obtain the character-level segmentation mask $\mathbf{C}$ with $|\mathcal{A}|$ channels, where $|\mathcal{A}|$ denote the size of alphabet $\mathcal{A}$. To this end, we obtain the layouts of keywords and the image generation process is introduced next.

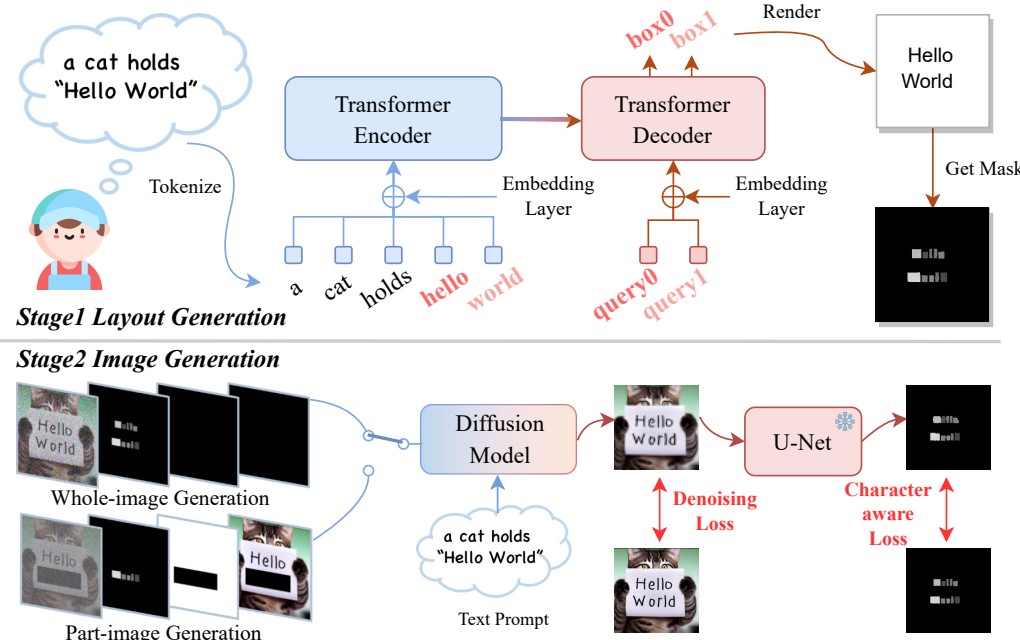

Figure 2: TextDiffuser consists of two stages. In the first Layout Generation stage, a Transformer-based encoder-decoder model generates character-level segmentation masks that indicate the layout of keywords in images from text prompts. In the second Image Generation stage, a diffusion model generates images conditioned on noisy features, segmentation masks, feature masks, and masked features (from left to right) along with text prompts. The feature masks can cover the entire or part of the image, corresponding to whole-image and part-image generation. The diffusion model learns to denoise features progressively with a denoising and character-aware loss. Please note that the diffusion model operates in the *latent space*, but we use the image pixels for better visualization.

## 3.2 Stage2: Image Generation

In this stage, we aim to generate the image guided by the segmentation masks $\mathbf{C}$ produced in the first stage. We use VAE [35] to encode the original image with shape $H \times W$ into 4-D latent space features $\mathbf{F} \in \mathbb{R}^{4 \times H' \times W'}$. Then we sample a time step $T \sim \text{Uniform}(0, T_{\max})$ and sample a Gaussian noise $\boldsymbol{\epsilon} \in \mathbb{R}^{4 \times H' \times W'}$ to corrupt the original feature, yielding $\hat{\mathbf{F}} = \sqrt{\bar{\alpha}_T}\mathbf{F} + \sqrt{1 - \bar{\alpha}_T}\boldsymbol{\epsilon}$ where $\bar{\alpha_T}$ is the coefficient of the diffusion process introduced in [25]. Also, we downsample the character-level segmentation mask $\mathbf{C}$ with three convolution layers, yielding 8-D $\hat{\mathbf{C}} \in \mathbb{R}^{8 \times H' \times W'}$. We also introduce two additional features, called 1-D feature mask $\hat{\mathbf{M}} \in \mathbb{R}^{1 \times H' \times W'}$ and 4-D masked feature $\hat{\mathbf{F}}_M \in \mathbb{R}^{4 \times H' \times W'}$. In the process of *whole-image generation*, $\hat{\mathbf{M}}$ is set to cover all regions of the feature and $\hat{\mathbf{F}}_M$ is the feature of a fully masked image. In the process of *part-image generation* (also called text inpainting), the feature mask $\hat{\mathbf{M}}$ represents the region where the user wants to generate, while the masked feature $\hat{\mathbf{F}}_M$ indicates the region that the user wants to preserve. To simultaneously train two branches, we use a masking strategy where a sample is fully masked with a probability of $\sigma$ and partially masked with a probability of $1 - \sigma$. We concatenate $\hat{\mathbf{F}}, \hat{\mathbf{C}}, \hat{\mathbf{M}}, \hat{\mathbf{F}}_M$ in the feature channel as a 17-D input and use denoising loss between the sampled noise $\boldsymbol{\epsilon}$ and the predicted noise $\boldsymbol{\epsilon}_\theta$:

$$l_{denoising} = ||\boldsymbol{\epsilon} - \boldsymbol{\epsilon}_\theta(\hat{\mathbf{F}}, \hat{\mathbf{C}}, \hat{\mathbf{M}}, \hat{\mathbf{F}}_M, \mathcal{P}, T)||_2^2. \tag{3}$$

Furthermore, we propose a character-aware loss to help the model focus more on text regions. In detail, we pre-train a U-Net [69] that can map latent features to character-level segmentation masks. During training, we fix its parameters and only use it to provide guidance by using a cross-entropy loss $l_{char}$ with weight $\lambda_{char}$ (See more details in Appendix A). Overall, the model is optimized with

$$l = l_{denoising} + \lambda_{char} * l_{char}. \tag{4}$$

Finally, the output features are fed into the VAE decoder to obtain the images.

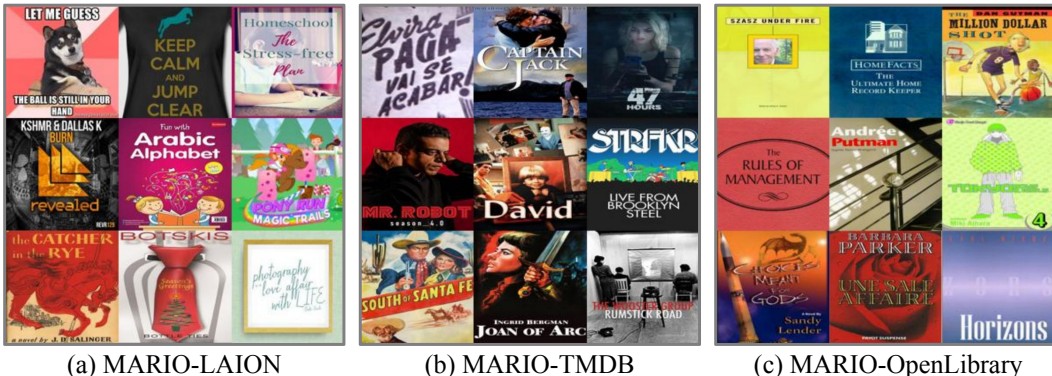

| (a) MARIO-LAION | (b) MARIO-TMDB | (c) MARIO-OpenLibrary |

Figure 3: Illustrations of three subsets of MARIO-10M. See more details in Appendix C.

## 3.3 Inference Stage

TextDiffuser provides a high degree of controllability and flexibility during inference in the following ways: (1) Generate images from user prompts. Notably, the user can modify the generated layout or edit the text to meet their personalized requirements; (2) The user can directly start from the second stage by providing a template image (*e.g.*, a scene image, handwritten image, or printed image), and a segmentation model is pre-trained to obtain the character-level segmentation masks (Appendix B); (3) Users can modify the text regions of a given image using text inpainting. Moreover, this operation can be performed multiple times. These experimental results will be presented in the next section.

## 4 MARIO Dataset and Benchmark

As there is no large-scale dataset designed explicitly for text rendering, to mitigate this issue, we collect 10 million image-text pairs with OCR annotations to construct the **MARIO-10M Dataset**. We further collect the **MARIO-Eval Benchmark** from the subset of the MARIO-10M test set and other existing sources to serve as a comprehensive tool for evaluating text rendering quality.

### 4.1 MARIO-10M Dataset

The **MARIO-10M** is a collection of about 10 million high-quality and diverse image-text pairs from various data sources such as natural images, posters, and book covers. Figure 3 illustrates some examples from the dataset. We design automatic schemes and strict filtering rules to construct annotations and clean noisy data (more details in Appendix D and Appendix E). The dataset contains comprehensive OCR annotations for each image, including text detection, recognition, and character-level segmentation annotations. Specifically, we use DB [42] for detection, PARSeq [4] for recognition, and manually train a U-Net [69] for segmentation. We analyze the performance of OCR tools in Appendix F. The total size of MARIO-10M is 10,061,720, from which we randomly chose 10,000,000 samples as the training set and 61,720 as the testing set. MARIO-10M is collected from three data sources:

**MARIO-LAION** derives from the large-scale datasets LAION-400M [75]. After filtering, we obtained 9,194,613 high-quality text images with corresponding captions. This dataset comprises a broad range of text images, including advertisements, notes, posters, covers, memes, logos, etc.

**MARIO-TMDB** derives from The Movie Database (TMDB), which is a community-built database for movies and TV shows with high-quality posters. We filter 343,423 English posters using the TMDB API out of 759,859 collected samples. Since each image has no off-the-shelf captions, we use prompt templates to construct the captions according to movie titles.

**MARIO-OpenLibrary** derives from Open Library, which is an open, editable library catalog that creates a web page for each published book. We first collect 6,352,989 original-size Open Library covers in bulk. Then, we obtained 523,684 higher-quality images after filtering. Like MARIO-TMDB, we manually construct captions using titles due to the lack of off-the-shelf captions.

## 4.2 MARIO-Eval Benchmark

The **MARIO-Eval benchmark** serves as a comprehensive tool for evaluating text rendering quality collected from the subset of the MARIO-10M test set and other sources. It comprises 5,414 prompts in total, including 21 prompts from DrawBenchText [72], 175 prompts from DrawTextCreative [46], 218 prompts from ChineseDrawText [49] and 5,000 image-text pairs from a subset of the MARIO-10M test set. The 5,000 image-text pairs are divided into three sets of 4,000, 500, and 500 pairs, and are named LAIONEval4000, TMDBEval500, and OpenLibraryEval500 based on their respective data sources. We offer examples in Appendix G to provide a clearer understanding of MARIO-Eval.

**Evaluation Criteria:** We evaluate text rendering quality with MARIO-Eval from four aspects: (1) **Fréchet Inception Distance (FID)** [24] compares the distribution of generated images with the distribution of real images. (2) **CLIPScore** calculates the cosine similarity between the image and text representations from CLIP [29, 60, 23]. (3) **OCR Evaluation** utilizes existing OCR tools to detect and recognize text regions in the generated images. Accuracy, Precision, Recall, and F-measure are metrics to evaluate whether keywords appear in the generated images. (4) **Human Evaluation** is conducted by inviting human evaluators to rate the text rendering quality of generated images using questionnaires. More explanations are shown in Appendix H.

# 5 Experiments

## 5.1 Implementation Details

For the *first* stage, we utilize the pre-trained CLIP [60] to obtain the embedding of given prompts. The number of Transformer layers $l$ is set to 2, and the dimension of latent space $d$ is set to 512. The maximum length of tokens $L$ is set to 77 following CLIP [60]. We leverage a commonly used font "Arial.ttf" and set the font size to 24 to obtain the width embedding and also use this font for rendering. The alphabet $\mathcal{A}$ comprises 95 characters, including 26 uppercase letters, 26 lowercase letters, 10 digits, 32 punctuation marks, and a space character. After tokenization, only the first subtoken is marked as the keyword when several subtokens exist for a word.

For the *second* stage, we implement the diffusion process using Hugging Face Diffusers [82] and load the checkpoint *"runwayml/stable-diffusion-v1-5"*. Notably, we only need to modify the input dimension of the input convolution layer (from 4 to 17), allowing our model to have a similar scale of parameters and computational time as the original model. In detail, the height $H$ and $W$ of input and output images are 512. For the diffusion process, the input is with spatial dimension $H' = 64$ and $W' = 64$. We set the batch size to 768 and trained the model for two epochs, taking four days using 8 Tesla V100 GPUs with 32GB memory. We use the AdamW optimizer [47] and set the learning rate to 1e-5. Additionally, we utilize gradient checkpoint [10] and xformers [39] for computational efficiency. During training, we follow [25] to set the maximum time step $T_{max}$ to 1,000, and the caption is dropped with a probability of 10% for classifier-free guidance [27]. When training the part-image generation branch, the detected text box is masked with a likelihood of 50%. We use 50 sampling steps during inference and classifier-free guidance with a scale of 7.5 following [67].

## 5.2 Ablation Studies

**Number of Transformer layers and the effectiveness of width embedding.** We conduct ablation studies on the number of Transformer layers and whether to use width embedding in the Layout Transformer. The results are shown in Table 1. All ablated models are trained on the training set of MARIO-10M and evaluated on its test set. Results show that adding width embedding improves performance, boosting IoU by 2.1%, 2.9%, and 0.3% when the number of Transformer layers $l$ is set to 1, 2, and 4, respectively. The optimal IoU is achieved using two Transformer layers and the width embedding is included. See more visualization results in Appendix I.

**Character-level segmentation masks provide explicit guidance for generating characters.** The character-level segmentation masks provide explicit guidance on the position and content of characters during the generation process of TextDiffuser. To validate the effectiveness of using character-level segmentation masks, we train ablated models without using the masks and show results in Appendix

Table 1: Ablation about Layout Transformer.

| #Layer | Width($\mathcal{P}$) | IoU↑ |
|---|---|---|
| 1 | - | 0.268 |
| | ✓ | 0.289 |
| 2 | - | 0.269 |
| | ✓ | **0.298** |
| 4 | - | 0.294 |
| | ✓ | 0.297 |

Table 2: Ablation on weight of character-aware loss.

| $\lambda_{char}$ | Acc↑ |
|---|---|
| 0 | 0.396 |
| 0.001 | 0.486 |
| 0.01 | **0.494** |
| 0.1 | 0.420 |
| 1 | 0.400 |

Table 3: Ablation on two-branch training ratio $\sigma$.

| ratio | Acc↑ / Det-F↑ / Spot-F↑ |
|---|---|
| 0 | 0.344 / 0.870 / 0.663 |
| 0.25 | **0.562** / 0.899 / 0.636 |
| 0.5 | 0.552 / 0.881 / **0.715** |
| 0.75 | 0.524 / **0.921** / 0.695 |
| 1 | 0.494 / 0.380 / 0.218 |

Table 4: The performance of text-to-image compared with existing methods. TextDiffuser performs the best regarding CLIPScore and OCR evaluation while achieving comparable performance on FID.

| Metrics | StableDiffusion [67] | ControlNet [102] | DeepFloyd [12] | TextDiffuser |
|---|---|---|---|---|
| FID↓ | 51.295 | 51.485 | **34.902** | 38.758 |
| CLIPScore↑ | 0.3015 | 0.3424 | 0.3267 | **0.3436** |
| OCR(Accuracy)↑ | 0.0003 | 0.2390 | 0.0262 | **0.5609** |
| OCR(Precision)↑ | 0.0173 | 0.5211 | 0.1450 | **0.7846** |
| OCR(Recall)↑ | 0.0280 | 0.6707 | 0.2245 | **0.7802** |
| OCR(F-measure)↑ | 0.0214 | 0.5865 | 0.1762 | **0.7824** |

J. The generated texts are inaccurate and not coherent with the background compared with texts generated with TextDiffuser, highlighting the importance of explicit guidance.

**The weight of character-aware loss.** The experimental results are demonstrated in Table 2, where we conduct experiments with $\lambda_{char}$ ranging from [0, 0.001, 0.01, 0.1, 1]. We utilize DrawBenchText [72] for evaluation and use Microsoft Read API to detect and recognize the texts in generated images. We use Accuracy (Acc) as the metric to justify whether the detected words *exactly match* the keywords. We observe that the optimal performance is achieved when $\lambda_{char}$ is set to 0.01, where the score is increased by 9.8% compared with the baseline ($\lambda_{char} = 0$).

**The training ratio of whole/part-image generation branches.** We explore the training ratio $\sigma$ ranging from [0, 0.25, 0.5, 0.75, 1] and show results in Table 3. When $\sigma$ is set to 1, it indicates that only the whole-image branch is trained and vice versa. We evaluate the model using DrawBenchText [72] for the whole-image generation branch. For the part-image generation branch, we randomly select 1,000 samples from the test set of MARIO-10M and randomly mask some of the detected text boxes. We utilize Microsoft Read API to detect and recognize the reconstructed text boxes in generated images while using the F-measure of text detection results and spotting results as metrics (denoted as Det-F and Spot-F, respectively). The results show that when the training ratio is set to 50%, the model performs better on average (0.716).

### 5.3 Experimental Results

**Quantitative Results.** For the *whole-image generation task*, we compare our method with Stable Diffusion (SD) [67], ControlNet [102], and DeepFloyd [12] in quantitative experiments with the publicly released codes and models detailed in Appendix K. DeepFloyd [12] uses two super-resolution modules to generate higher resolution $1024 \times 1024$ images compared with $512 \times 512$ images generated by other methods. We use the Canny map of printed text images generated with our first stage model as conditions for ControlNet [102]. Please note that we are not able to compare with Imagen [72], eDiff-i [2], and GlyphDraw [49] due to the lack of open-source code, checkpoints or APIs. According to Table 4, we demonstrate the quantitative results of the text-to-image task compared with existing methods. Our TextDiffuser obtains the best CLIPScore while achieving comparable performance in terms of FID. Besides, TextDiffuser achieves the best performance regarding four OCR-related metrics. TextDiffuser outperforms those methods without explicit text-related guidance by a large

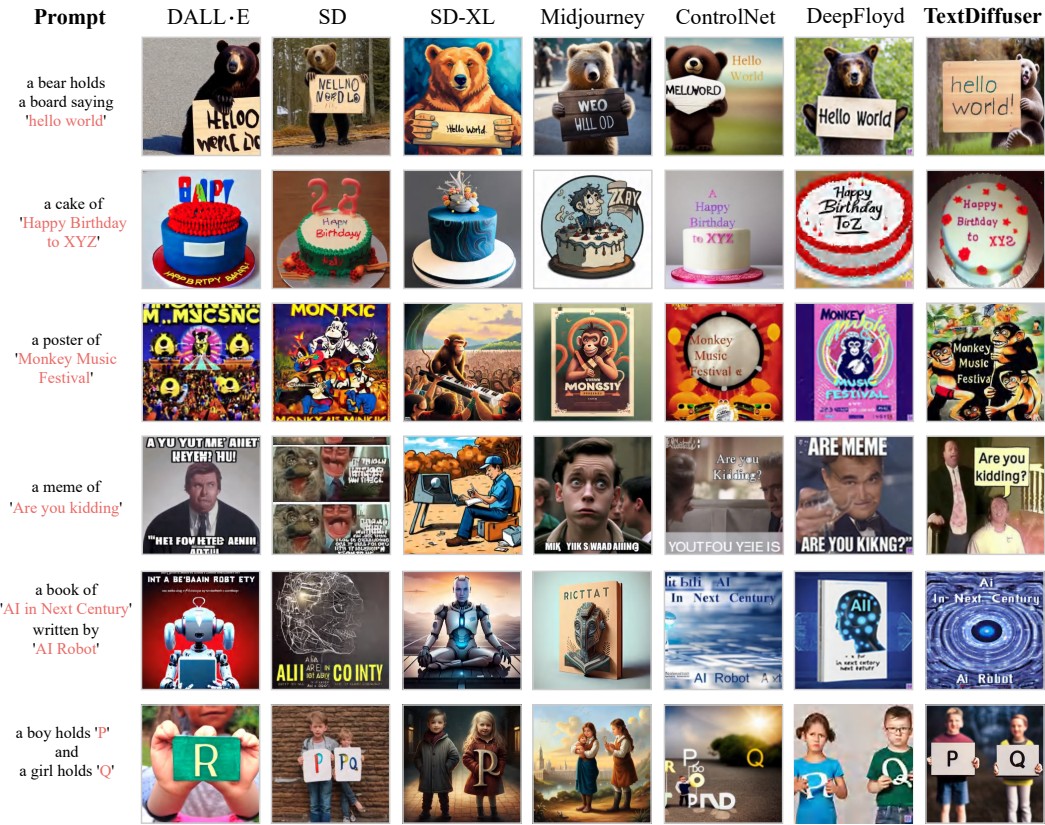

| Prompt | DALL·E | SD | SD-XL | Midjourney | ControlNet | DeepFloyd | **TextDiffuser** |
|---|---|---|---|---|---|---|---|

Figure 4: Visualizations of whole-image generation compared with existing methods. The first three cases are generated from prompts and the last three cases are from given printed template images.

margin (*e.g.*, 76.10% and 60.62% better than Stable Diffusion and DeepFloyd regarding F-measure), highlighting the significance of explicit guidance. As for the ***part-image generation task***, we cannot evaluate our method since no methods are specifically designed for this task to our knowledge.

**Qualitative Results.**    For the ***whole-image generation task***, we further compare with closed-source DALL·E [63], Stable Diffusion XL (SD-XL), and Midjourney by showing qualitative examples generated with their official API services detailed in Appendix K. Figure 4 shows some images generated from prompts or printed text images by different methods. Notably, our method generates more readable texts, which are also coherent with generated backgrounds. On the contrary, although the images generated by SD-XL and Midjourney are visually appealing, some generated text does not contain the desired text or contains illegible characters with incorrect strokes. The results also show that despite the strong supervision signals provided to ControlNet, it still struggles to generate images with accurate text consistent with the background. We also initiate a comparison with the Character-Aware Model [46] and the concurrent work GlyphDraw [49] using samples from their papers as their open-source code, checkpoints or APIs are not available. Figure 5 shows that TextDiffuser performs better than these methods. For instance, the Character-Aware Model suffers from misspelling issues (*e.g.*, 'm' in 'Chimpanzees') due to its lack of explicit control, and GlyphDraw struggles with rendering images containing multiple text lines. For the ***part-image generation task***, we visualize some results in Figure 6. In contrast to text editing tasks [88], we give the model sufficient flexibility to generate texts with reasonable styles. For instance, the image in the second row and first column contains the word "country" in green, while the model generates the word "country" in yellow. This is reasonable since it follows the style of the nearest word "range". Besides, our method can render realistic text coherent with the background, even in complex cases such as clothing. More qualitative results are shown in Appendix L.

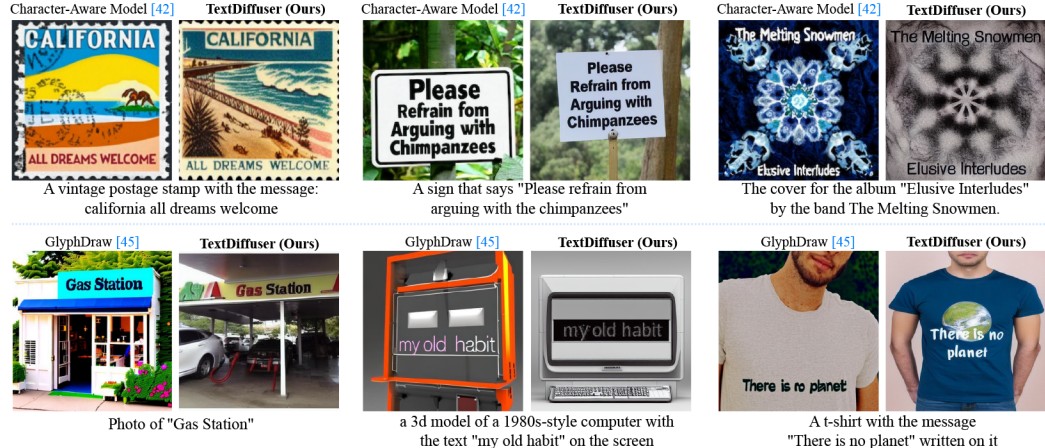

Figure 5: Comparison with Character-Aware Model [46] and the concurrent GlyphDraw [49].

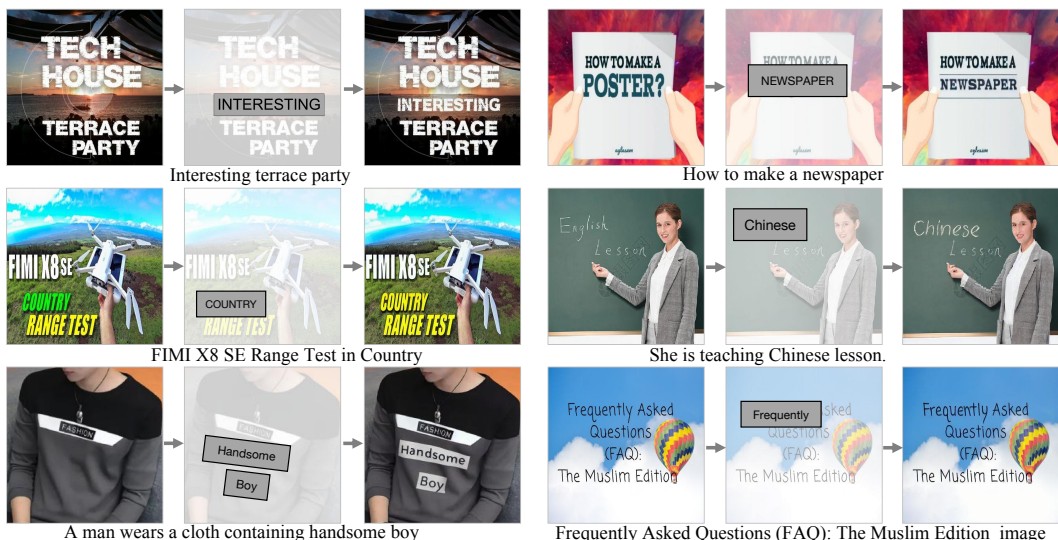

Figure 6: Visualizations of part-image generation (text inpainting) from given images.

**User Studies.** For the ***whole-image generation task***, the designed questionnaire consists of 15 cases, each of which includes two multiple-choice questions: (1) Which of the following images has the best text rendering quality? (2) Which of the following images best matches the text description? For the ***part-image generation task***, the questionnaire consists of 15 cases, each of which includes two rating questions: (1) How is the text rendering quality? (2) Does the drawn text harmonize with the unmasked region? The rating scores range from 1 to 4, and 4 indicates the best. Overall, we have collected 30 questionnaires, and the results are shown in Figure 8. We can draw two conclusions: (1) The generation performance of TextDiffuser is significantly better than existing methods. (2) Users are satisfied with the inpainting results in most cases. More details are shown in Appendix M.

**Time and Parameter Efficiency** For the time efficiency, the first stage of Layout Generation leverages an auto-regressive Transformer whose prediction time correlates with the number of keywords. Specifically, we conduct experiments to evaluate the time overhead for different numbers of keywords, including 1 (1.07±0.03s), 2 (1.12±0.09s), 4 (1.23±0.13s), 8 (1.57±0.12s), 16 (1.83±0.12s), and 32 (1.95±0.28s). Meanwhile, the second stage of image generation is independent of the number of queries (7.12±0.77s). For the parameter efficiency, TextDiffuser builds upon Stable Diffusion 1.5 (859M parameters), adding a Layout Transformer in the first stage (+25M parameters) and modifying the second stage (+0.75M parameters), augmenting it by only about 3% in terms of parameters.

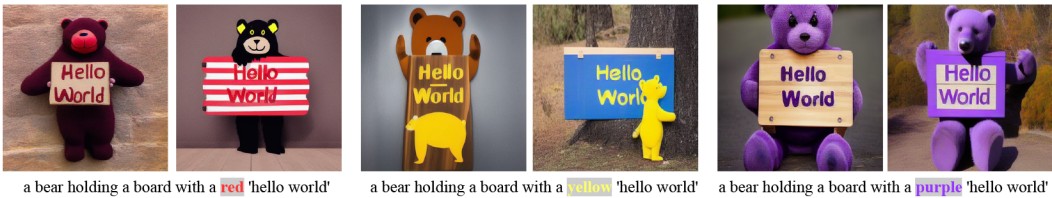

a bear holding a board with a red 'hello world'    a bear holding a board with a yellow 'hello world'    a bear holding a board with a purple 'hello world'

Figure 7: Demonstration of using language descriptions to control text color.

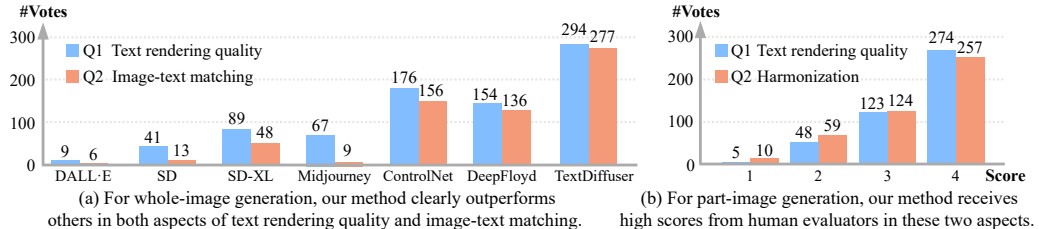

(a) For whole-image generation, our method clearly outperforms others in both aspects of text rendering quality and image-text matching.

(b) For part-image generation, our method receives high scores from human evaluators in these two aspects.

Figure 8: User studies for whole-image generation and part-image generation tasks.

**Text Color Controllability**  In Figure 7, we showcase TextDiffuser's capability in controlling the color of generated texts through language descriptions. The visualization results show that TextDiffuser can successfully control the color of rendered text, further enhancing its controllability.

## 6  Discussion and Conclusion

**Discussion.** We show that TextDiffuser maintains the capability and generality to create general images without text rendering in Appendix N. Besides, we compare our method with a text editing model in Appendix O, showing that TextDiffuser generates images with better diversity. We also present the potential of TextDiffuser on the text removal task in Appendix P. As for the **limitations and failure cases**, TextDiffuser uses the VAE networks to encode images into low-dimensional latent spaces for computational efficiency following latent diffusion models [67, 49, 2], which has a limitation in reconstructing images with small characters as shown in Appendix Q. We also observed failure cases when generating images from long text and showed them in Appendix Q. As for the **broader impact**, TextDiffuser can be applied to many designing tasks, such as creating posters and book covers. Additionally, the text inpainting task can be used for secondary creation in many applications, such as Midjourney. However, there may be some ethical concerns, such as the misuse of text inpainting for forging documents. Therefore, techniques for detecting text-related tampering [86] need to be applied to enhance security. **In conclusion,** we propose a two-stage diffusion model called TextDiffuser to generate images with visual-pleasing texts coherent with backgrounds. Using segmentation masks as guidance, the proposed TextDiffuser shows high flexibility and controllability in the generation process. We propose MARIO-10M containing 10 million image-text pairs with OCR annotations. Extensive experiments and user studies validate that our method performs better than existing methods on the proposed benchmark MARIO-Eval. **For future work,** we aim to address the limitation of generating small characters by using OCR priors following OCR-VQGAN [66] and enhance TextDiffuser's capabilities to generate images with text in multiple languages. **Disclaimer** Please note that the model presented in this paper is intended for academic and research purposes **ONLY**. Any use of the model for generating inappropriate content is strictly prohibited and is not endorsed by this paper. The responsibility for any misuse or improper use of the model lies solely with the users who generated such content, and this paper shall not be held liable for any such use.

## 7  Acknowledgement

This research was supported by the Research Grant Council of the Hong Kong Special Administrative Region under grant number 16203122.

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

# Appendix

## A  Architecture of U-Net and Design of Character-Aware Loss

As shown in Figure 9, the U-Net contains four downsampling operations and four upsampling operations. The input will be downsampled to a maximum of 1/16. To provide the character-aware loss, the input feature $\mathbf{F}$ is 4-D with spatial size $64 \times 64$, while the output is 96-D (the length of alphabet $\mathcal{A}$ plus a null symbol indicating the non-character pixel) also with spatial size $64 \times 64$. Subsequently, a cross-entropy loss is calculated between the output feature (need to convert the predicted noise into predicted features) and the resized $64 \times 64$ character-level segmentation mask $\mathbf{C}'$. The U-Net is pre-trained using the training set of MARIO-10M for one epoch. We utilize the Adadelta optimizer [97] and set the learning rate to 1. When training the diffusion model, the U-Net is frozen and only used to provide character-aware guidance.

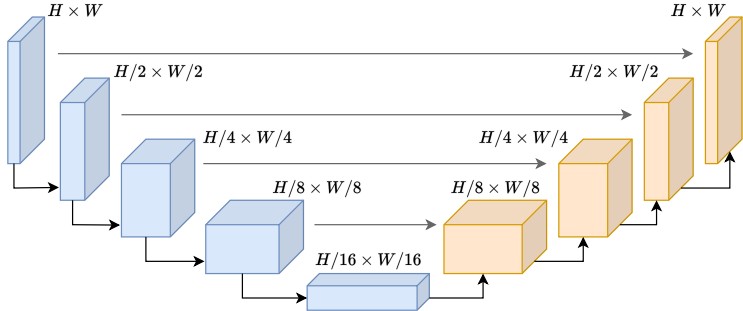

Figure 9: The architecture of U-Net contains four downsampling and upsampling operations.

## B  Character-Level Segmentation Model

We train the character-level segmentation model based on U-Net, whose architecture is similar to the architecture shown in Figure 9. We set the input size to $256 \times 256$, ensuring that most characters are readable at this resolution. We train the segmentation model using synthesized scene text images [19], printed text images, and handwritten text images[3], totaling about 4M samples. We employ data augmentation strategies (*e.g.*, blurring, rotation, and color enhancement) to make the segmentation model more robust. The segmentation model is trained for ten epochs using the Adadelta optimizer [97] with a learning rate of 1. Figure 10 shows some samples in the training dataset.

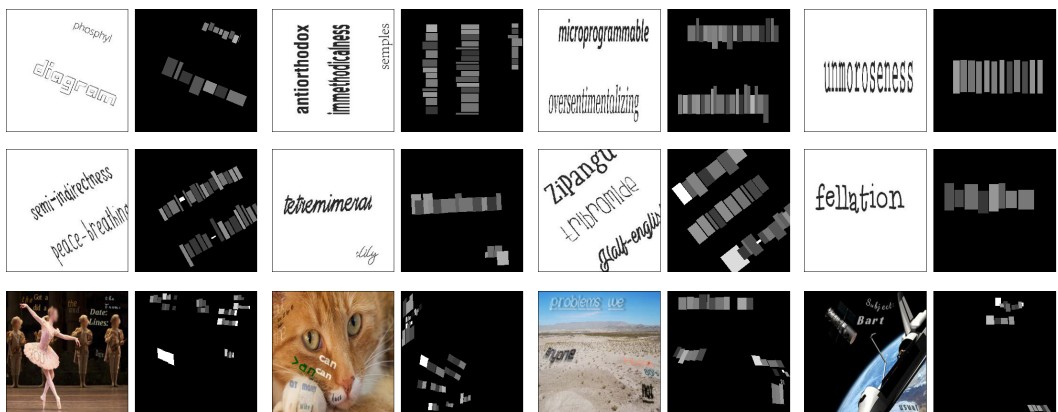

Figure 10: Visualization of some training samples for the character-level segmentation model. The top, middle, and bottom roles are samples from printed, handwritten, and scene datasets.

---

[3]https://github.com/Belval/TextRecognitionDataGenerator

## C More Details in MARIO-10M

Table 5: Number of texts per image in MARIO-10M.

| #Words | 1 | 2 | 3 | 4 | 5 | 6 | 7 | 8 |
|---|---|---|---|---|---|---|---|---|
| #Images | 592,153 | 1,148,481 | 1,508,185 | 1,610,056 | 1,549,852 | 1,430,750 | 1,229,714 | 930,809 |
| #Ratio | 5.9% | 11.5% | 15.1% | 16.1% | 15.5% | 14.3% | 12.3% | 9.3% |

More samples are shown in Figure 11. The number of texts per image in MARIO-10M is shown in Table 5. Also, the MARIO-10M dataset reveals that about 90% of the text regions maintain a horizontal orientation with rotation angles smaller than 5 degrees without perspective changes. Hence, our layout generation model is designed to predict horizontal bounding boxes by detecting the coordinates of their left-top and bottom-right points. Adapting our model to predict more realistic scene text is feasible by detecting enhanced coordinates, such as eight coordinates for four points.

## D MARIO-10M Caption Templates

Since TMDB movie/TV posters and Open Library book covers have no off-the-shelf captions, we construct them based on their titles with the following templates. {XXX} is a placeholder for title.

For **MARIO-TMDB**:

- Logo {XXX}
- Text {XXX}
- Title {XXX}
- Title text {XXX}
- A poster with a title text of {XXX}
- A poster design with a title text of {XXX}
- A quality movie print with a title text of {XXX}
- A film poster of {XXX}
- A movie poster of {XXX}
- A movie poster titled {XXX}
- A movie poster named {XXX}
- A movie poster with text {XXX} on it
- A movie poster with logo {XXX} on it
- A movie poster with a title text of {XXX}
- An illustration of {XXX} movie
- An photography of {XXX} movie
- A TV show poster titled {XXX}
- A TV show poster of {XXX}
- A TV show poster with logo {XXX} on it
- A TV show poster with a title text of {XXX}
- A TV show poster with text {XXX}
- A TV show poster named {XXX}

For **MARIO-OpenLibrary**:

- A book with a title text of {XXX}
- A book design with a title text of {XXX}
- A book cover with a title text of {XXX}
- A book of {XXX}
- A cover named {XXX}
- A cover titled {XXX}
- A book with text {XXX} on it
- A book cover with logo {XXX} on it

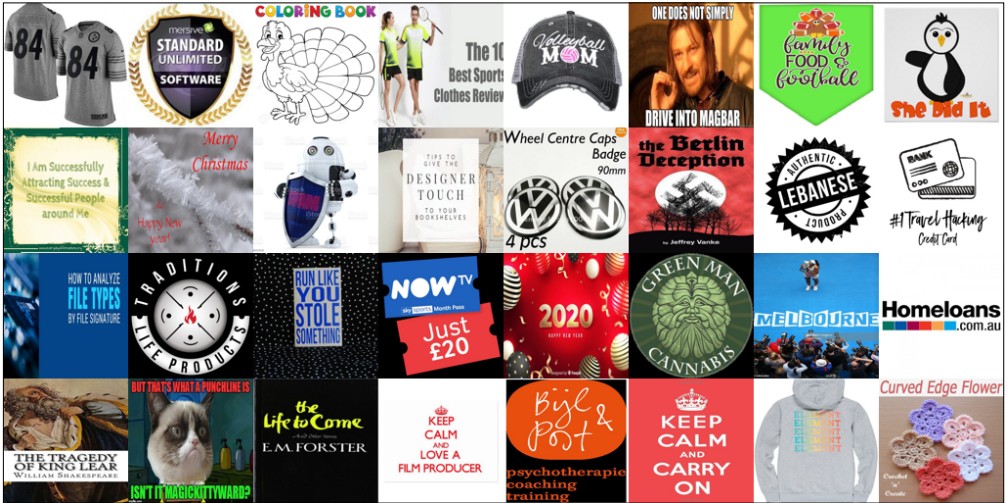

(a) Samples in MAION-LAION

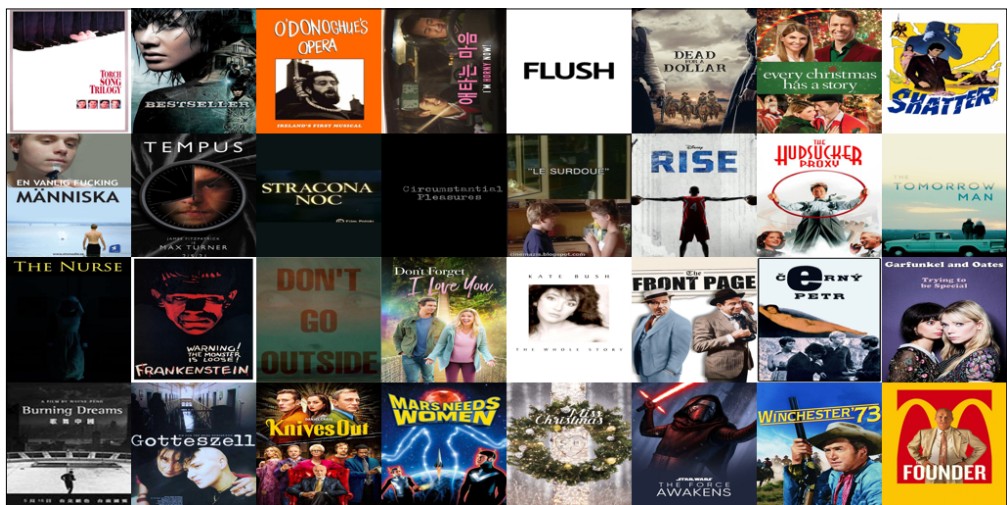

(b) Samples in MARIO-TMDB

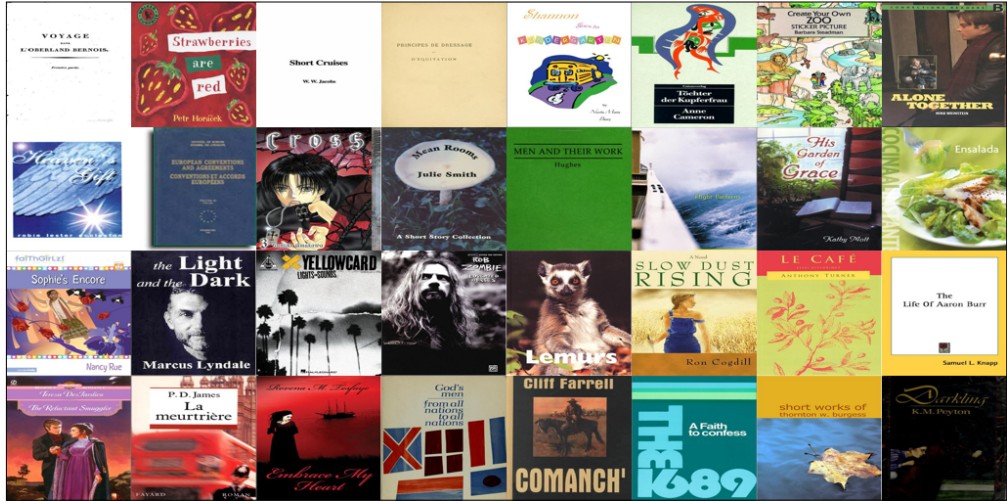

(c) Samples in MARIO-OpenLibrary

Figure 11: More samples in MARIO-10M.

# E  MARIO-10M Filtering Rules

We clean data with five strict **filtering rules** to obtain high-quality data with text:

- **Height and width are larger than 256**. Low-resolution samples often contain illegible texts, negatively impacting the training process.

- **Will not trigger NSFW**. For the MARIO-LAION subset, we filter out those samples triggering the "not sure for work" flag to mitigate ethical concerns.

- **The number of detected text boxes should be within** [**1,8**]. We detect texts with DB [42]. Samples with too many texts typically have small areas for each text, which makes them difficult to recognize. Therefore, we remove these samples from the dataset.

- **Text areas are more than 10% of the whole image**. According to Appendix B, we train a UNet [69] using SynthText [19] to obtain character-level segmentation masks of each sample. This criterion ensures that the text regions will not be too small.

- **At least one detected text appears in the caption**. Noticing that the original dataset contains many noisy samples, we add this constraint to increase the relevance between images and captions. We utilize PARSeq [4] for text recognition.

# F    Analysis of OCR Performance on MARIO-10M

As we rely on OCR tools to annotate MARIO-10M, it is necessary to evaluate the performance of these tools. Specifically, we manually annotate 100 samples for text recognition, detection, and character-level segmentation masks, then compare them with the annotations given by OCR tools. The results are shown in Table 6. We notice that the performance of existing methods is lower than their results on text detection and spotting benchmarks. Taking DB [42] as an example, it can achieve text detection 91.8% precision on ICDAR 2015 dataset [34] while only achieving 76% on MARIO-10M. This is because there are many challenging cases in MARIO-10M, such as blurry and small text. Besides, a domain gap may exist since DB is trained on scene text detection datasets, while MARIO-10M comprises text images in various scenarios. Future work may explore more advanced recognition, detection, and segmentation models to mitigate the noise in OCR annotations. We demonstrate some OCR results in Figure 12.

Table 6: OCR performance on MARIO-10M. *IOU (binary)* means we treat each pixel as two classes: characters and non-characters. The evaluation of recognition is included in the spotting task.

| Detection | | | Spotting | | | Segmentation | |
|---|---|---|---|---|---|---|---|
| Precision | Recall | F-measure | Precision | Recall | F-measure | IOU (binary) | IOU |
| 0.76 | 0.79 | 0.78 | 0.73 | 0.75 | 0.74 | 0.70 | 0.59 |

Figure 12: Visualization of some OCR annotations in MARIO-10M.

# G    Samples in MARIO-Eval

Table 7: Details of each subset in MARIO-Eval.

| Subset | Size | Off-the-shelf Captions | GT Images |
|---|---|---|---|
| LAIONEval4000 | 4,000 | ✓ | ✓ |
| TMDBEval500 | 500 | ✗ | ✓ |
| OpenLibraryEval500 | 500 | ✗ | ✓ |
| DrawBenchText [72] | 21 | ✓ | ✗ |
| DrawTextCreative [46] | 175 | ✓ | ✗ |
| ChineseDrawText [49] | 218 | ✓ | ✗ |

As illustrated in Table 7, MARIO-Eval contains 5,414 prompts with six subsets. The ground truth images of some samples are shown in Figure 13, and captions for each category are shown below:

**LAIONEval4000**:

- 'Royal Green' Wristband Set
- Are 'Digital Nomads' in Trouble Well Not Exactly
- 'Sniper Elite' One Way Trip A Novel Audiobook by 'Scott' McEwen Thomas Koloniar Narrated by Brian Hutchison
- 'Travel Artin' Logo
- Falls the 'Shadow' Welsh Princes 2

**TMDBEval500**:

- A movie poster with text 'Heat' on it
- A poster design with a title text of 'Deadpool 2'
- A TV show poster named 'Ira Finkelstein s Christmas'
- A movie poster with logo 'Playing for Change Songs Around The World Part 2 ' on it
- A movie poster titled 'Dreams of a Land'

**OpenLibraryEval500**:

- A book cover with a title text of 'On The Apparel Of Women'
- A book with text 'Precalculus' on it
- A book design with a title text of 'Dream master nightmare'
- A book cover with logo 'Thre Poetical Works of Constance Naden' on it
- A book cover with a title text of 'Discovery'

**DrawBenchText**:

- A storefront with 'Hello World' written on it.
- A storefront with 'Text to Image' written on it.
- A sign that says 'Diffusion'.
- A sign that says 'NeurIPS'.
- New York Skyline with 'Google Research Pizza Cafe' written with fireworks on the sky.

**DrawTextCreative**:

- a grumpy sunflower with a 'no solar panels' sign
- A photo of a rabbit sipping coffee and reading a book. The book title 'The Adventures of Peter Rabbit' is visible.

- a graffiti art of the text 'free the pink' on a wall
- A professionally designed logo for a bakery called 'Just What I Kneaded'.
- scholarly elephant reading a newspaper with the headline 'elephants take over the world'

**ChineseDrawText**:

- A street sign on the street reads 'Heaven rewards those who work hard'
- There is a book on the table with the title 'Girl in the Garden'
- Kitten holding a sign that reads 'I want fish'
- A robot writes 'Machine Learning' on a podium
- In a hospital, a sign that says 'Do Not Disturb'

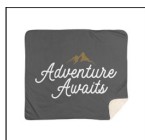 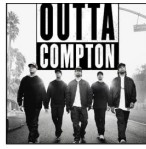 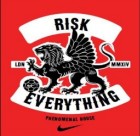 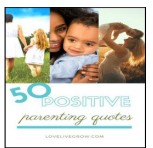 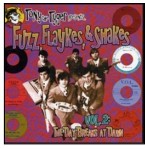

'Adventure Awaits' Home Blanket by Katie Rose s Artist Shop — Win 2 Tickets To An Advanced Screening Of Straight 'Outta Compton' In Atlanta Courtesy Of HHS1987 Aug 11th — 'Risk Everything' — '50 positive parenting quotes' gentle parenting peaceful parenting gentle discipline — V A TONY THE TYGVOLUME TWO THE DAY BREAKS AT DAWN LP

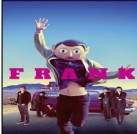 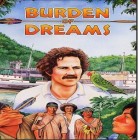 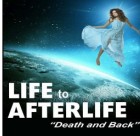 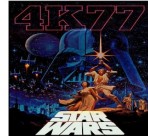 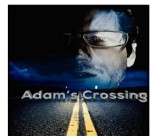

A poster with a title text of 'Frank' — A TV show poster of 'Burden of Dreams' — A poster with a title text of 'Life to Afterlife Death and Back' — A movie poster with a title text of 'Star Wars' — A movie poster with logo 'Adams Crossing' on it

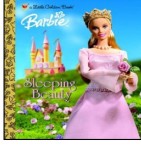 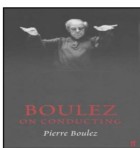 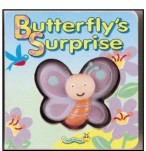 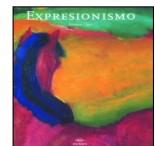 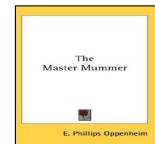

A cover titled 'Sleeping Beauty' — A cover titled 'Boulez on Conducting' — A book design with a title text of 'Butterflys Surprise' — A book with a title text of 'Expresionismo' — A book with text 'The master mummer' on it

Figure 13: We demonstrate five samples for LAIONEval4000 (top), TMDBEval500 (middle), and OpenLibrary500 (bottom).

## H  Implementation Details of Evaluation Criteria

To evaluate the performance of TextDiffuser quantitatively, we utilize three criteria, including FID, CLIPScore, and OCR Evaluation. We detail the calculation of each criterion below.

**FID.** We calculate the FID score using the pytorch-fid repository. Please note that the proposed MARIO-Eval benchmark's three subsets (DrawTextCreative, DrawBenchText, and ChineseDrawText) do not contain ground truth images. Therefore, we utilize 5,000 images in the other three subsets (LAIONEval4000, TMDBEval500, OpenLibraryEval500) as the ground truth images. We calculate the FID score using the 5,414 generated images and the 5,000 ground truth images.

**CLIP Score.** We calculate the CLIP score using the clipscore repository. However, as with the FID score, we cannot calculate the CLIP score for the DrawTextCreative, DrawBenchText, and ChineseDrawText subsets due to the lack of ground truth images. Therefore, we only calculate the score on LAIONEval4000, TMDBEval500, and OpenLibraryEval500 subsets and report the average CLIP score.

**OCR Evaluation.** For the MARIO-Eval benchmark, we use quotation marks to indicate the keywords that need to be painted on the image. Taking the caption [A cat holds a paper saying 'Hello World'] as an example, the keywords are 'Hello' and 'World'. We then use Microsoft Read API to detect and recognize text in the image. We evaluate OCR performance using accuracy, precision, recall, and F-measure. If the detected text matches the keywords exactly, it is considered correct. Precision represents the proportion of detected text that matches the keywords, while recall represents the proportion of keywords that appear in the image. We report the mean values of precision and recall, and calculate the F-measure using the following formula:

$$\text{F-measure} = \frac{2 \times \text{Precision} \times \text{Recall}}{\text{Precision} + \text{Recall}}. \tag{5}$$

## I  Visualization of Layouts Generated by Layout Transformer

We visualize some generated layouts in Figure 14, showing that the Transformer can produce reasonable layouts.

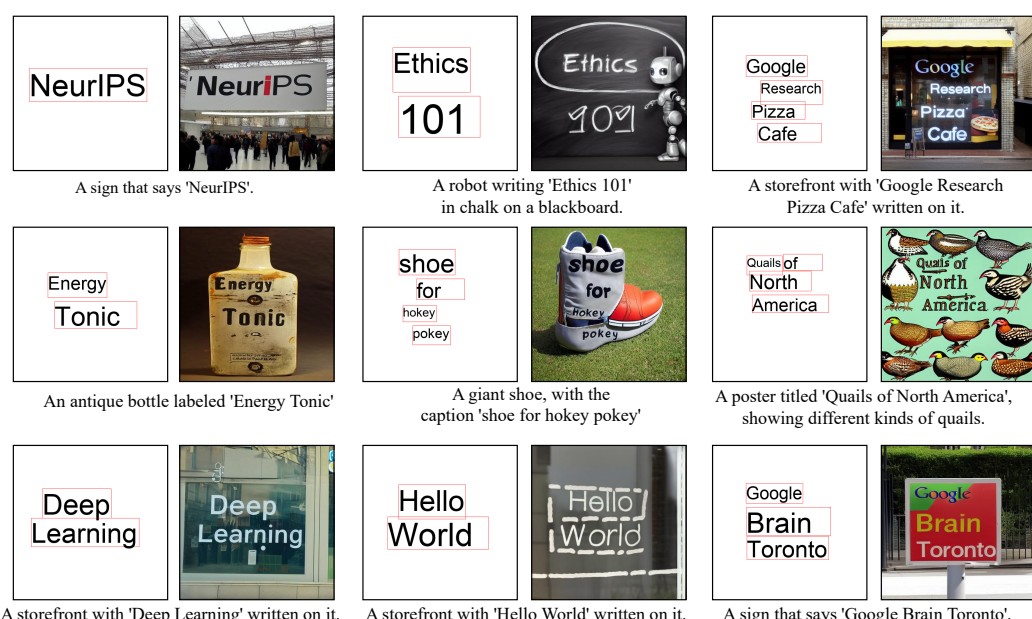

A sign that says 'NeurIPS'.

A robot writing 'Ethics 101' in chalk on a blackboard.

A storefront with 'Google Research Pizza Cafe' written on it.

An antique bottle labeled 'Energy Tonic'

A giant shoe, with the caption 'shoe for hokey pokey'

A poster titled 'Quails of North America', showing different kinds of quails.

A storefront with 'Deep Learning' written on it.

A storefront with 'Hello World' written on it.

A sign that says 'Google Brain Toronto'.

Figure 14: Visualization of the generated layouts and images.

## J   Experiment without Explicit Guidance of Segmentation Masks

As shown in Figure 15, we try to explore the generation without explicit guidance. For example, according to the first row, we set the value of character pixels to 1 and non-character pixels to 0 (*i.e.*, remove the content and only provide the position guidance). We observe that the model can generate some words similar to keywords but contain some grammatical errors (*e.g.*, a missing "l" in "Hello"). Further, according to the second row, we train TextDiffuser without segmentation masks (*i.e.*, remove both position and content guidance). In this case, the experiment is equivalent to directly fine-tuning a pre-trained latent diffusion model on the MARIO-10M dataset. The results show that the text rendering quality worsens, demonstrating explicit guidance's significance.

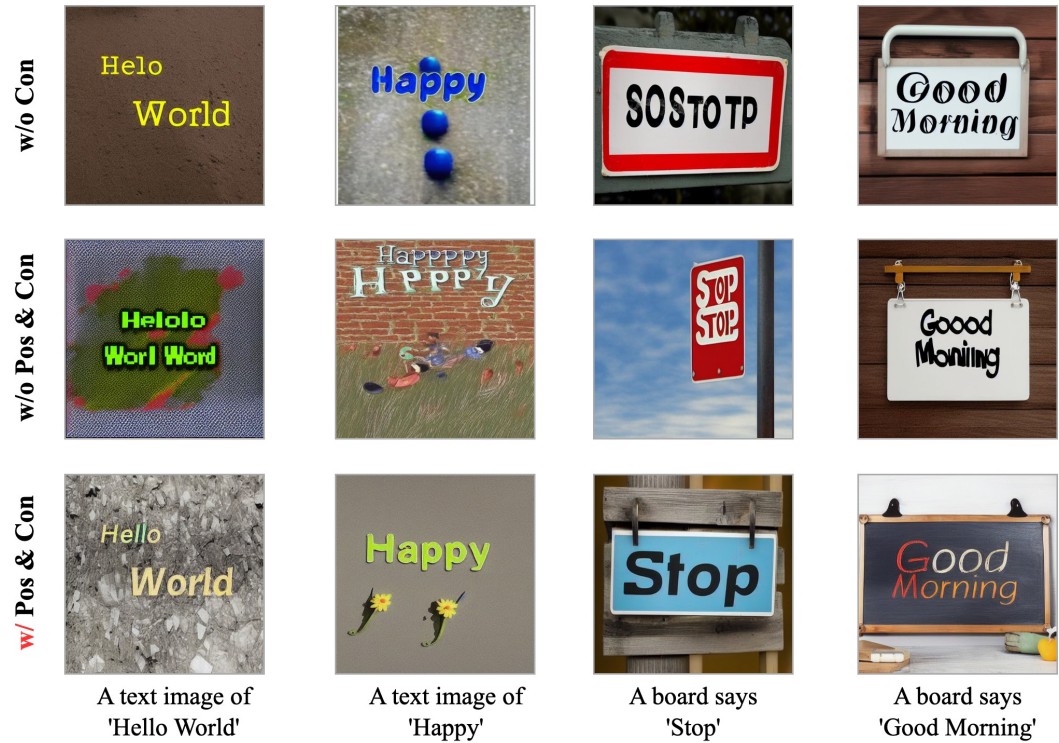

Figure 15: Visualization of generation without explicit guidance.

# K    Baseline Methods Experimental Settings

We introduced all baseline methods and their experimental settings when we used to compare them with the TextDiffuser as follows.

**DALL·E** [62] utilizes a text encoder to map a given prompt into a corresponding representation space. A prior model is then employed to map the text encoding to an image encoding. Finally, an image decoder generates an image based on the image encoding. Since there is no available code and model, we obtain the results using the provided API[4].

**Stable Diffusion (SD)** utilizes CLIP [60] text encoder to obtain the embedding of user prompts, pre-trained VAE to encode original images and conducts the diffusion process in the latent space for computation efficiency. We use the public pre-trained model *"runwayml/stable-diffusion-v1-5"* based on Hugging Face diffusers [82]. The number of sampling steps is 50, and the scale of classifier-free guidance is 7.5.

**Stable Diffusion XL (SD-XL)** is an upgraded version of SD, featuring more parameters and utilizing a more powerful language model. Consequently, it can be expected to better understand prompts compared to SD. As the source code and model are not publicly available, we obtained the generation results through a web API[5].

**Midjourney**[6] is a commercial project that runs on Discord, allowing users to interact with a bot via the command-line interface. We generated images using the default parameters of Midjourney. For example, we can generate an image using the following command: /imagine an image of 'hello world' in Midjourney.

**ControlNet** [102] aims to control diffusion models by adding conditions using zero-convolution layers. We use the public pre-trained model *"lllyasviel/sd-controlnet-canny"* released by ControltNet authors and the implementation from Hugging Face *diffusers* [82]. For fair comparisons, we use the printed text images generated by our first-stage model to generate Canny maps as the condition of ControlNet. We use default parameters for inference, where the low and high thresholds of canny map generation are set to 100 and 200, respectively. The number of inference steps is 20, and the scale of classifier-free guidance is 7.5.

**DeepFloyd** [12] designs three cascaded pixel-based diffusion modules to generate images of increasing resolution: 64x64, 256x256, and 1024x1024. All stage modules use frozen text encoders based on T5 Transformer [12]. Compared with CLIP [60], the T5 Transformer is a powerful language model that enables more effective text understanding. We use the public pretrained models released by DeepFloyd authors and the implementation from Hugging Face diffusers [82]. We use default models and parameters for inference, where the three pretrained cascaded models are *"DeepFloyd/IF-I-XL-v1.0"*, *"DeepFloyd/IF-II-L-v1.0"*, and *"stabilityai/stable-diffusion-x4-upscaler"*.

---

[4]https://openai.com/product/dall-e-2
[5]https://beta.dreamstudio.ai/generate
[6]https://www.midjourney.com/

# L    Visualization of More Generation Results by Our TextDiffuser

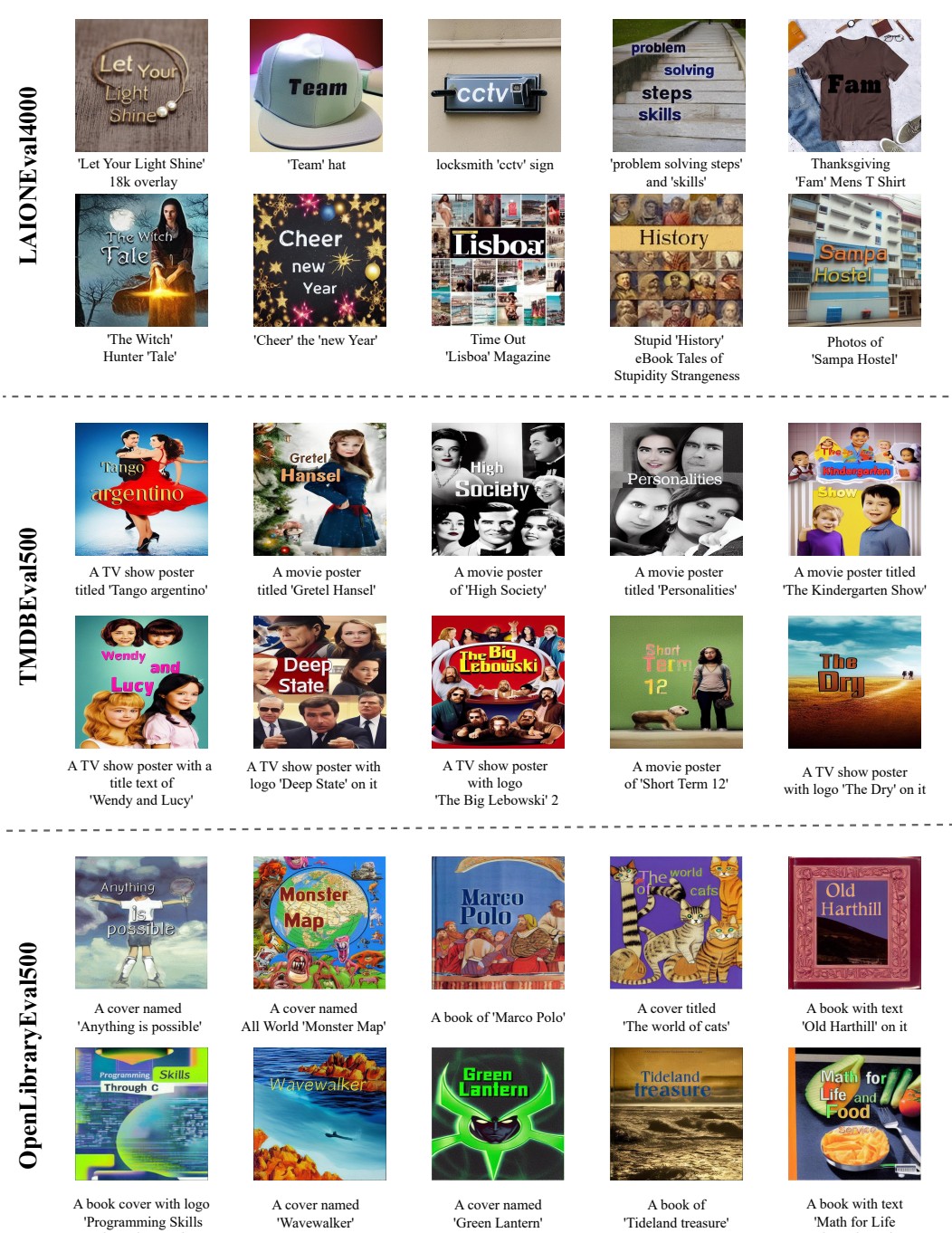

Figure 16: Visualization of more generation results by our TextDiffuser in LAIONEval4000, TMD-BEval500, and OpenLibrary500.

**DrawTextCreative**

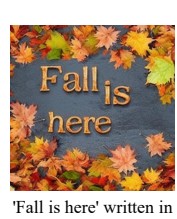

a giant shoe,
with the caption
'shoe for hokey pokey'

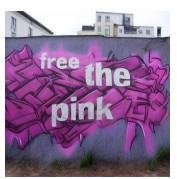

'Fall is here' written in
autumn leaves
floating on a lake

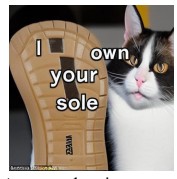

a graffiti art of the text
'free the pink' on a wall

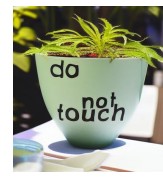

A meme showing a cat
attacking a shoe, with the
message 'I own your shoe'

plant in a fancy pot with
a 'do not touch' sign

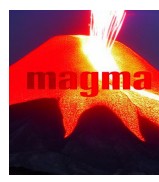

A t-shirt with the
message
'There is no planet B'

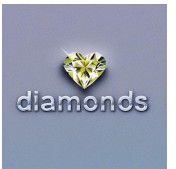

pillow in the shape
of words
'ready for the weekend'

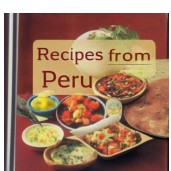

a volcano erupting,
with the text
'magma' in red

a logo for the company
'diamonds', with a diamon
in the shape of a heart

A large recipe book
titled
'Recipes from Peru'.

**DrawBenchText**

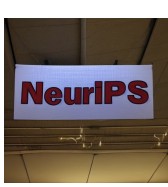

A sign that
says 'NeurIPS'.

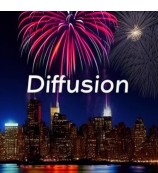

New York Skyline with
'Diffusion' written
with fireworks on the sky.

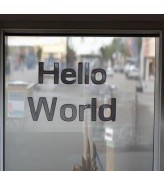

A storefront with
'Hello World'
written on it.

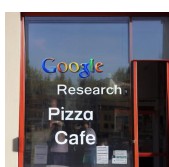

A storefront with
'Google Research Pizza Cafe'
written on it.

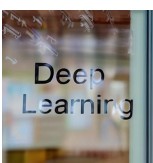

A storefront with
'Deep Learning'
written on it.

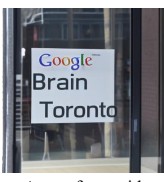

A storefront with
'Google Brain Toronto'
written on it.

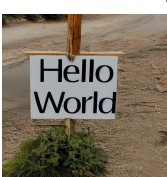

A sign that says
'Hello World'.

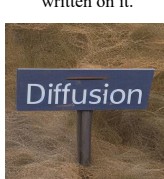

A sign that says
'Diffusion'.

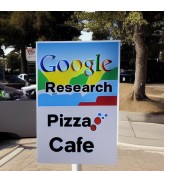

A sign that says
'Google Research
Pizza Cafe'.

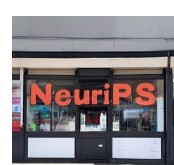

A storefront with
'NeurIPS' written on it.

**ChineseDrawText**

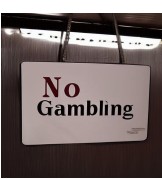

Prohibition sign
"No Gambling"
hung on the entrance
of the casino

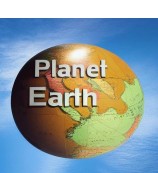

A globe with the words
"Planet Earth"
written in bold
letters with continents
in bright colors

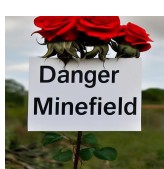

A photo of roses
surrounded by
a sign in the distance
that reads
"Danger Minefield"

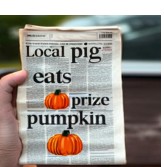

A newspaper headline read
"Local pig eats
prize pumpkin"
and a photo showed
a half-eaten pumpkin

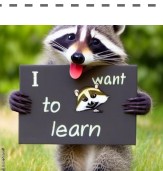

Little raccoon
holding
a sign that reads
"I want to learn"

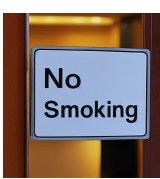

A "No Smoking" sign
is placed in the hotel

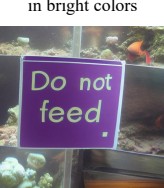

A sign saying
"Do not feed"
in the aquarium

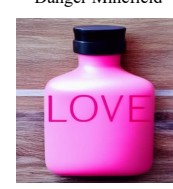

A pink bottle
that says "LOVE"

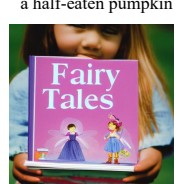

A little girl is holding a
book with the words
"Fairy Tales" in her hands

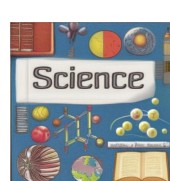

Books with the word
"Science" printed
on them

Figure 17: Visualization of more generation results by our TextDiffuser in DrawTextCreative, DrawBenchText, and ChineseDrawText.

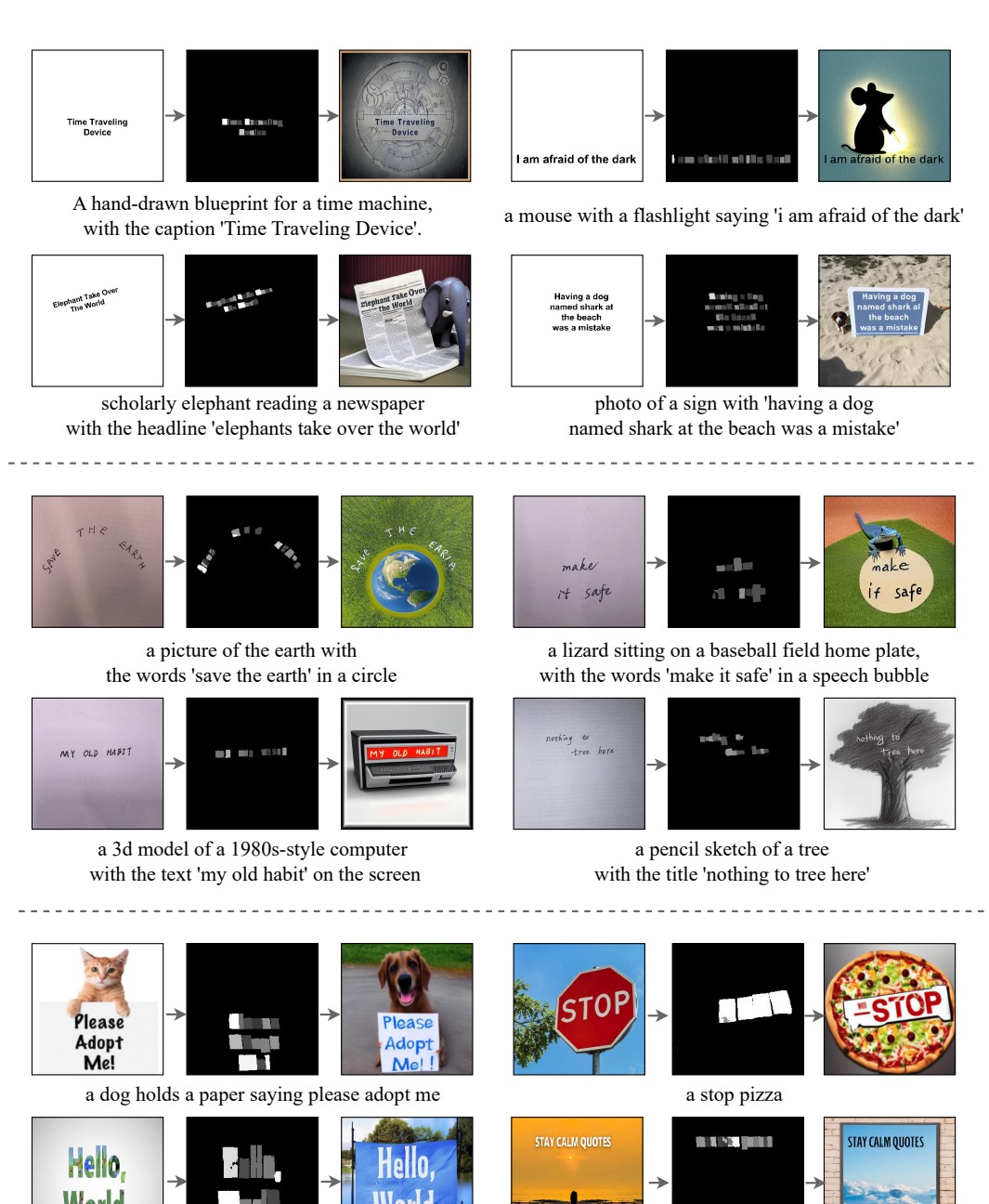

Figure 18: Visualization of more generation results by our TextDiffuser for the text-to-image with template task.

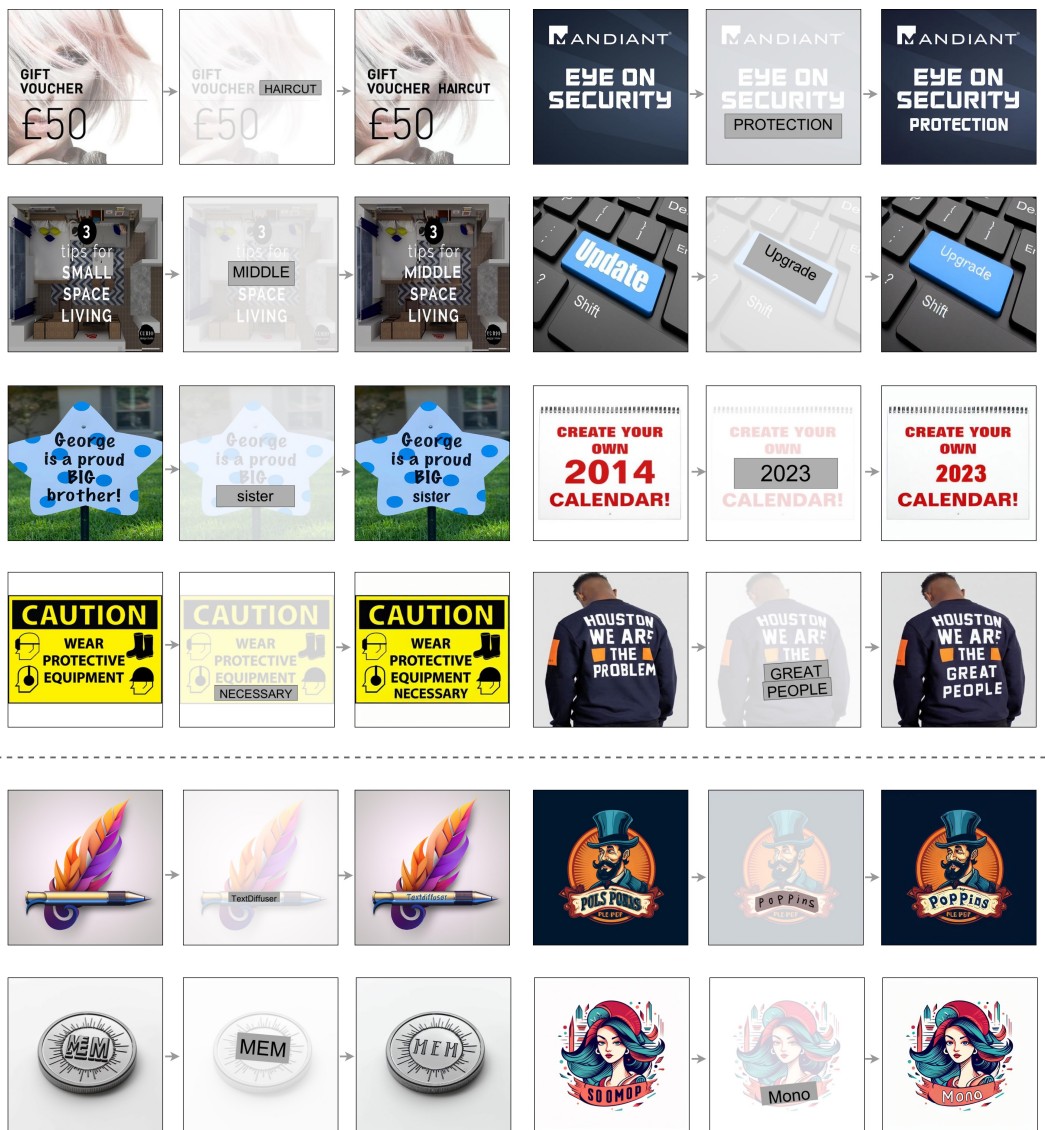

Figure 19: Visualization of more generation results for text inpainting. The images above the dash lines are from the test set of MARIO-10M, while the images below the dash lines are collected from the Midjourney community.

# M  More Details about User Study

**User study on the whole-image generation task.**   The questionnaire consists of 15 cases, each of which includes two multiple-choice questions:

- Which of the following images has the best text rendering quality?
- Which of the following images best matches the text description?

In particular, the first question focuses on the text rendering quality. Taking Figure 20 as an example[7], we expect the model to render the word "EcoGrow" accurately (*i.e.*, without any missing or additional characters). The second question, on the other hand, focuses on whether the overall image matches the given prompt. For example, in Figure 20 (G), although the generated text is correct, it fails to meet the requirement in the prompt that the letter looks like a plant. We instruct users to select the best option. In cases where multiple good options are difficult to distinguish, they could choose multiple options. If users are unsatisfied with the options, they could decide not to select any.

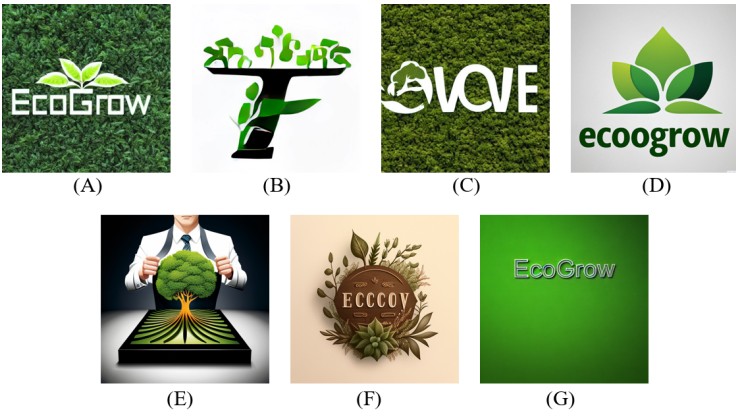

[(8/15)] A logo for the company 'EcoGrow', where the letters look like plants.

Figure 20: One case in the user study for the whole-image generation task.

**User study on the part-image generation task.**   We aim to let users vote on the quality of text inpainting (from 4 to 1, the higher, the better). We also designed two questions:

- How is the text rendering quality?
- Does the drawn text harmonize with the unmasked region?

Specifically, the first question concentrates on the accuracy of the text. The second question focuses on whether the generated part is harmonious with the unmasked part (*i.e.*, whether the background and texture are consistent).

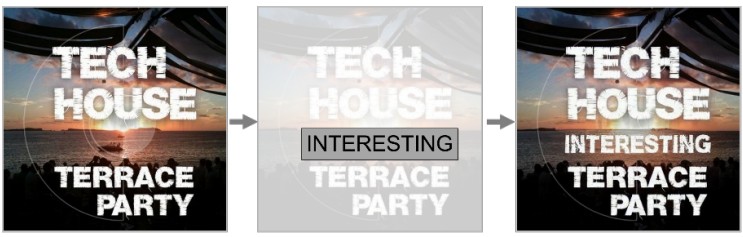

Figure 21: One case in the user study for the part-image generation task.

---

[7](A) **TextDiffuser**; (B) DALL·E; (C) SD; (D) IF; (E) SD-XL; (F) Midjourney; (G) ControlNet.

# N    Generating Images without Text

To show the generality of TextDiffuser, we experiment with generating images that do not contain texts and show results in Figure 22. Although TextDiffuser is fine-tuned with MARIO-10M, it still maintains a good generation ability for generating general images. Therefore, users have more options when using TextDiffuser, demonstrating its flexibility. We also provide quantitative evaluations to demonstrate TextDiffuser's generality in generating non-text general images. We compare TextDiffuser with our baseline Stable Diffusion 1.5 as they have the same backbone. For a quantitative evaluation, the FID scores of 5,000 images generated by prompts randomly sampled from MSCOCO are as in Table 8. The results indicate that TextDiffuser can maintain the ability to generate natural images even after fine-tuning the domain-specific dataset.

Table 8: FID scores on MSCOCO compared with Stable Diffusion.

| Sampling Steps | Stable Diffusion | TextDiffuser |
| --- | --- | --- |
| 50 | 26.47 | 27.72 |
| 100 | 27.02 | 27.04 |

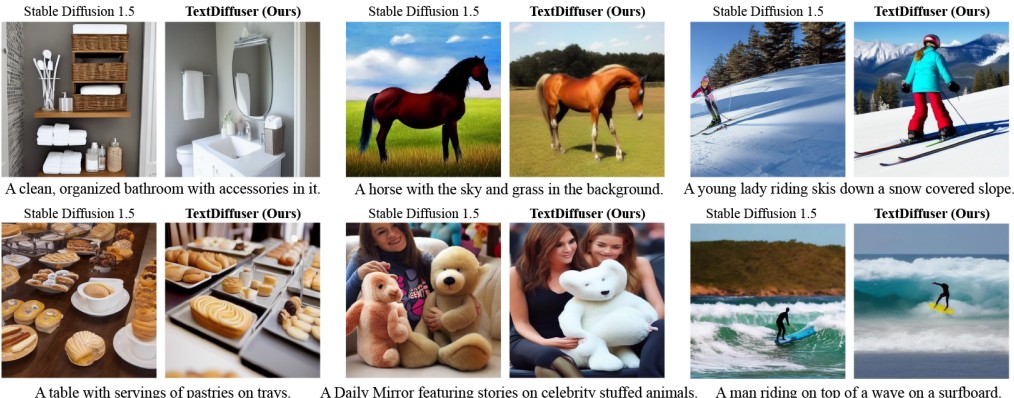

Figure 22: Visualizations of general images generated by Stable Diffusion 1.5 and TextDiffuser.

## O    Comparisons between TextDiffuser and a Text Editing Model

We visualize some results in Figure 23 compared with a text editing model SRNet [88]. Please note that the introduced text inpainting task differs from the text editing task in three aspects: (1) The text editing task usually relies on the synthesized text image dataset for training (synthesizing two images with the different text given a background image and font as pairs). In contrast, the text inpainting task follows the mask-and-recover training scheme and can be trained with any text images. (2) Text editing emphasizes the preservation of the original fonts, while text inpainting allows for greater freedom. For example, we conduct four samplings for each case, and the generated results exhibit diversity and present reasonable font styles. (3) Text editing tasks cannot add text, highlighting the significance of the introduced text inpainting task.

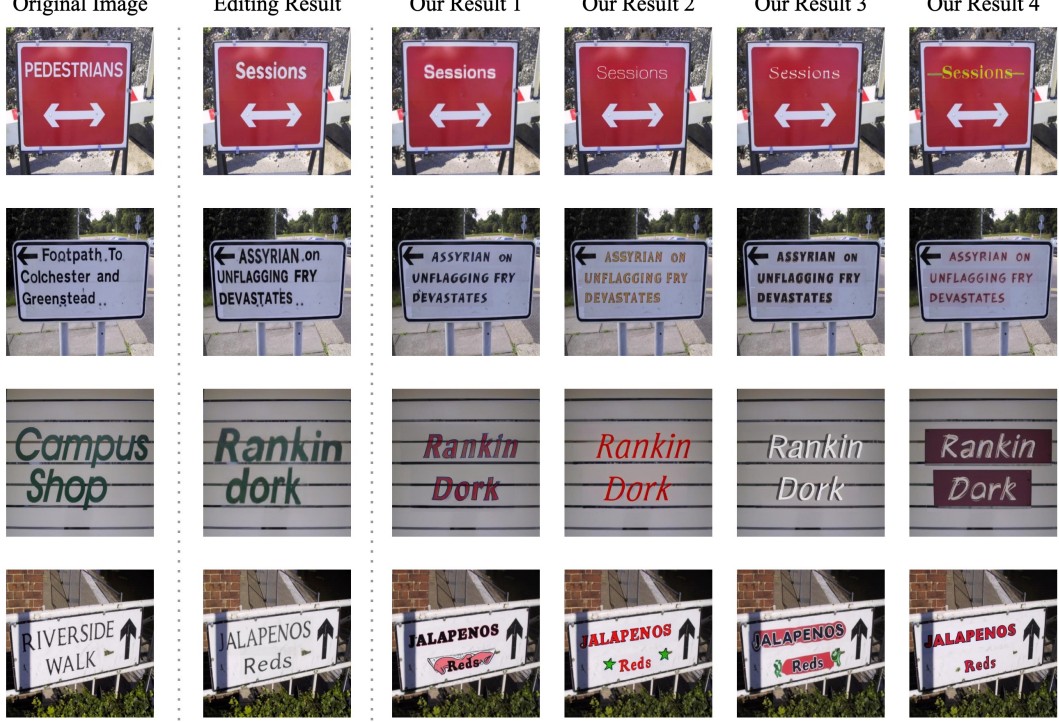

Figure 23: Comparison with text editing model. Four cases are obtained from the paper of SRNet.

# P  Experimental Results of Text Removal

We demonstrate some results of text removal in Figure 24, and the cases are obtained from the paper of EraseNet [44]. We can easily transform the text inpainting task into a text removal task by providing a mask and setting all regions to non-character in the segmentation mask. Experimental results demonstrate that our method can achieve results similar to the ground truth.

Original Image          Ground Truth          Inpainting Mask          Our Result

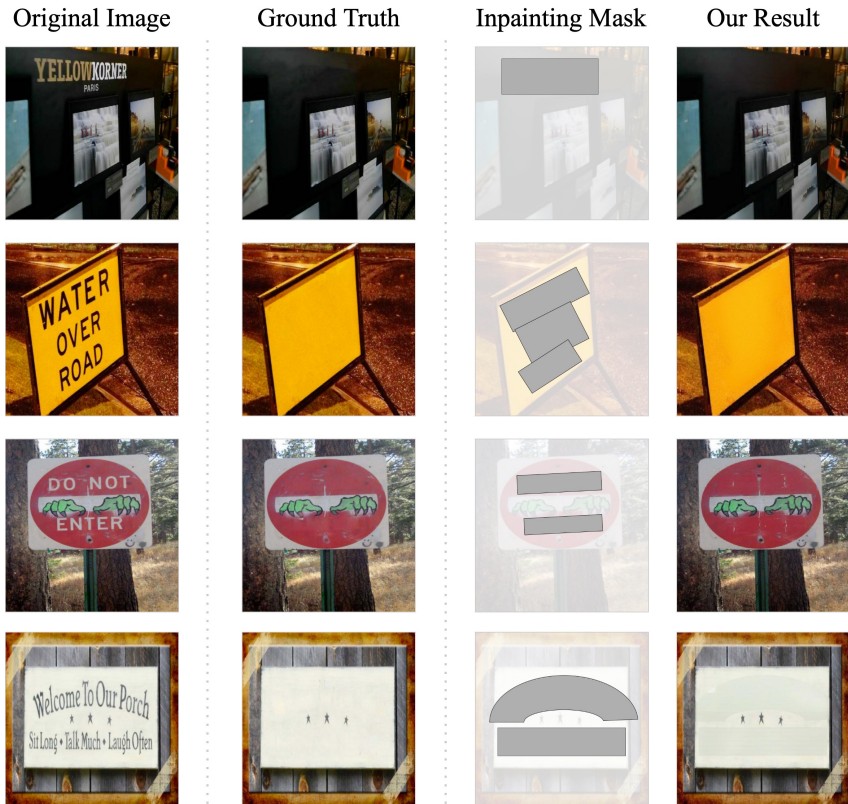

Figure 24: Experimental results on text removal. Four cases are obtained from the paper of EraseNet.

# Q  Limitations and Failure Cases

We observe failure cases when generating images with small characters and from long text.

**Generating images with small characters.**   TextDiffuser uses the VAE networks to encode images into low-dimensional latent spaces for computational efficiency following latent diffusion models [67, 49, 2]. However, the compression process can result in losing details when generating images with small characters. As illustrated in Figure 25, we observe that the VAE fails to reconstruct small characters, where reconstructed strokes are unclear and reduce the legibility of the text. According to the generated images, the small characters appear to have vague or disjointed strokes (*e.g.*, the character 'l' in 'World' and character 'r' in 'Morning'), which could impact the readability. As shown in Figure 26, we notice that using a more powerful backbone, such as Stable Diffusion 2.1, can mitigate this issue. When the image resolution is enhanced from $512\times512$ to $768\times768$ using Stable Diffusion 2.1 (instead of 1.5), the latent space resolution also increases from $64\times64$ to $96\times96$, enhancing the character-level representation. As the cost, the inference latency rises from 8.5s to 12.0s with a batch size of 1. Therefore, how to render small characters while maintaining the same time cost is worth further study.

**Generating images from long text.**   We observed failure cases when generating images from long text with many keywords, where the generated words in the layouts are disordered and overlapped,

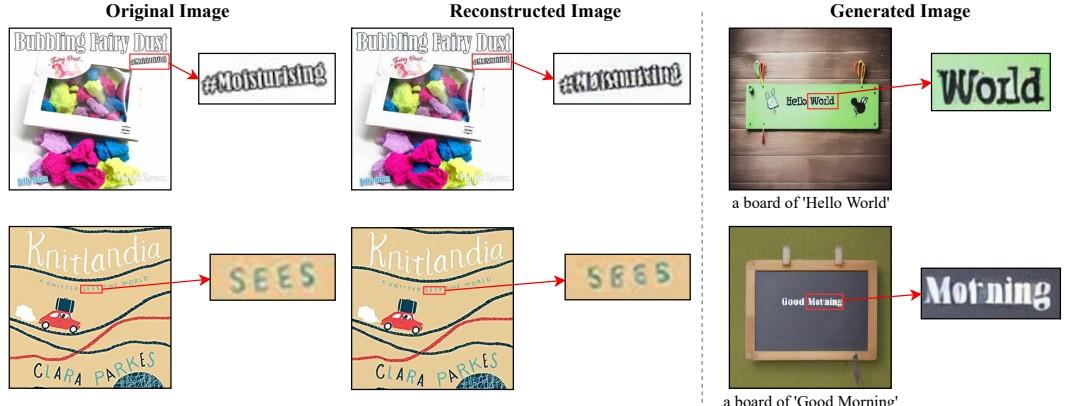

Figure 25: The issue of generating images with small characters.

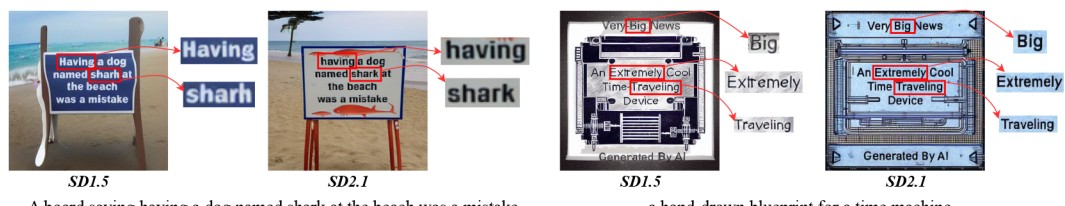

Figure 26: Pre-trained on high-resolution Stable Diffusion 2.1 enhances the legibility of small text.

as shown in Figure 27. One possible reason could be that training examples containing numerous keywords tend to have more noise (*i.e.*, those images usually contain dense and small text), leading to a higher likelihood of detection and recognition errors. To address this, we could consider enhancing the capability of OCR tools in the future to mitigate the noise.

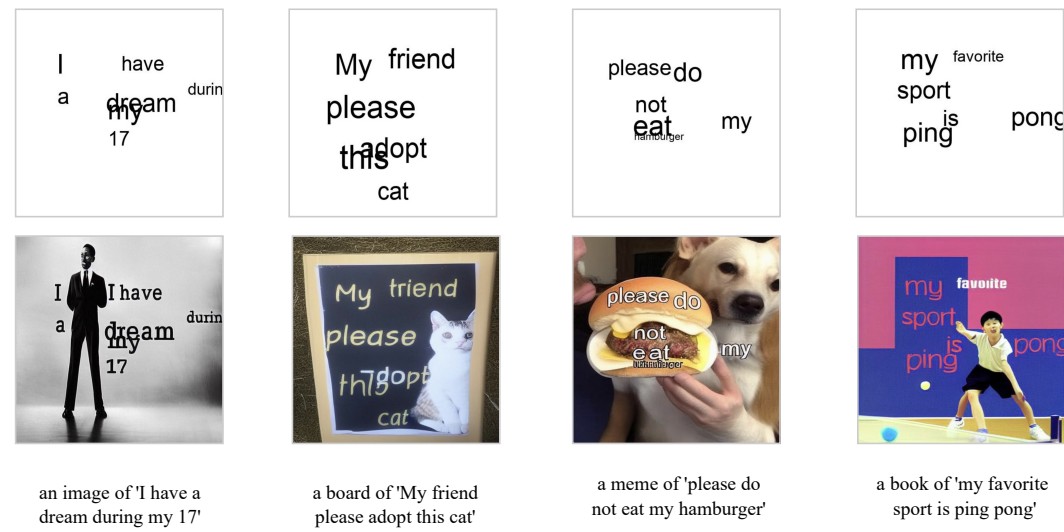

Figure 27: The issue of dealing with a large number of keywords.

