# OpenReview forum: "TextDiffuser: Diffusion Models as Text Painters"
_NeurIPS.cc/2023/Conference — NeurIPS 2023 poster_

### Official Review · Reviewer_AzyM · 2023-07-03

**Soundness:** 3 good
**Presentation:** 3 good
**Contribution:** 3 good
**Rating:** 3
**Confidence:** 5

**Summary:**

This paper proposed TextDiffuser to achieve accurate text rendering for diffusion models. TextDiffuser generates character-level text layouts to guide the text rendering process (image generation). The model is evaluated on 3 tasks including text-to-image, text-to-image with template, and text inpainting to demonstrate its flexibility and controllability. Additionally, the paper contributes the first large-scale text image dataset with OCR annotations, named MARIO-10M.

**Strengths:**

1.	The paper is well motivated, which tries to solve the problem of generating text image with diffusion model. Also, the paper is well organized and easy to follow.
2.	The TextDiffuser model is flexible, which can be adaptive with different conditions as shown in the experiments.
3.	A new dataset specifically designed for text rendering is proposed.


**Weaknesses:**

1.	Mismatch of character level layout and the real text. The character level layout is generated from standard fonts (rendered using toolkit such as opencv), however, the real character can be of any style or font, which leads to a mismatch of character level layout. In another word, guided by such a layout, the generated text will tend to have the same character interval and aspect ratio of the standard font, which means it will limit the generation diversity of fonts.
2.	Generation diversity has not been considered. There exist several reasonable text styles for the same background image. As the proposed model cannot explicit control the attribute of texts such as color, layout, font, style, etc., it is important to measure the generation diversity of the model.
3.	The performance comparison seems to be limited to general text-to-image diffusion models with no specific optimization on text painting. A comparison with some state-of-the-art text generation methods is necessary to show the quality of text generation.


**Questions:**

1. How does the model perform on more complex characters such as Chinese characters? A simple layout guidance seems to only constrain the position and order of letters, without any specific improvement on the quality of generation details. I wonder if there is any method to improve the adaptiveness on other language?

**Limitations:**

Limitations and potential impacts have been discussed.

---

> ### Author Rebuttal · Authors · 2023-08-09
>
> Thank you for your review and feedback. We aim to address and clarify each point raised.
>
> **Character-Level Layout and Real Character Style:**
> The primary role of our character-level layout is to inform and guide the position and content of visual text, without restricting it to specific styles or fonts. This design choice empowers TextDiffuser to dynamically adapt to varied text styles, as the model is trained on the diverse MARIO-10M dataset. To illustrate this, Figure 6 in Appendix I of our supplementary material shows the model interprets a standard font layout and coherently translates it into diverse fonts and styles according to the context.
>
> **Generation Diversity:**
> To clarify, TextDiffuser can explicitly control text layouts by conditioning the second stage of image generation on the first stage's prediction or by user-provided layout templates. As outlined in Line 278 and visualized in Figure 15 of Appendix O, even with the same layout input, TextDiffuser exhibits diversity in text attributes such as color, font and style. Furthermore, Figure(b) in the attached PDF provides additional demonstrations of TextDiffuser's diversity and control capability, like color modulation via prompts. We will emphasize this diversity and capability more clearly in the paper.
>
> **Performance Comparison:**
> To the best of our knowledge, state-of-the-art text generation methods like Imagen [67], eDiff-i [2], Character-Aware Model [42], and GlyphDraw [45] have not publicly released open-source code, checkpoints or APIs for throughout comparisom, as highlighted in Lines 240-241.
> Still, we initiated a comparison with Character-Aware Model [42] and the concurrent work GlyphDraw [45] using samples from their papers. In Figure(d) of the attached PDF, TextDiffuser performs better than these methods. For instance, [42] suffers from misspell issues (e.g., ‘m’ in ‘Chimpanzees’) due to its lack of explicit control and [45] struggles with rendering images containing multiple text lines. We will release our code to facilitate future comparisons.
>
> **Adaptability to Complex Characters, e.g., Chinese:**
> TextDiffuser is promising to handle complex characters like Chinese characters by collecting a Chinese version of the MARIO dataset and training a character-level segmentation network for Chinese characters. We highlight the effectiveness of our proposed explicit layout guidance in the quality of generation details due to its strong constraint not only on the position but also the content of each characters. The OCR evaluations in Table 4 quantitatively validate the accuracy of our text rendering. Concurrently, our qualitative experiment in Appendix J shows the drastic degradation in quality without explicit guidance. We believe our design of explicit guidance will benefit more complex characters generation.
>
> We believe that our contributions of comprehensive model, dataset and benchmark are significant for future research and development in this field. We will release code, model, and dataset to advance the research community and broaden the applications of our approach.

---

> > ### Comment · Reviewer_AzyM · 2023-08-16
> >
> > The author did not reply to my question and the problems still exist.

---

> > > ### Author Response · Authors · 2023-08-17
> > > **Detailed Explanation of the Question**
> > >
> > > Thank you for reviewing and responding. Here is a more detailed answer to your question:
> > >
> > > **Question**
> > > > How does the model perform on more complex characters such as Chinese characters? A simple layout guidance seems to only constrain the position and order of letters, without any specific improvement on the quality of generation details. I wonder if there is any method to improve the adaptiveness on other language?
> > >
> > > **Answer**
> > > Currently, TextDiffuser cannot render text in other languages. TextDiffuser is specialized for English text generation due to its training on English-centric datasets and embeddings of segmentation masks. We agree that layout guidance mainly focuses on layout constraints. We can improve the adaptiveness of TextDiffuser for other languages from two aspects:
> > >
> > > * **Supporting Multilingual Text Generation**:
> > >     - Modeling: Extend the embeddings of segmentation masks and the vocabulary of the text encoder tokenizer to accommodate other languages.
> > >     - Data: Use a Multilingual MARIO dataset for training.
> > > * **Improving Complex Character Generation Details**:
> > >     - Fine-grained Control Signals: Use fine-grained segmentation masks to explicitly guide glyphs.
> > >     - High-Resolution Rendering: Adopt more advanced frameworks like Stable Diffusion 2.1 and DeepFloyd with high-resolution rendering capabilities to improve the generation details of complex characters.
> > >
> > > We appreciate your feedback and remain open to any further questions.

---

> > > ### Comment · Area_Chair_h1uG · 2023-08-20
> > >
> > > Hi,
> > >
> > > Could you please comment more on whether your concerns have been addressed after reading the author's new reply?
> > >
> > > AC

---

> > > > ### Comment · Reviewer_AzyM · 2023-08-21
> > > >
> > > > The author did not give any experimental comparisons in the response. So, my major concerns remain unsolved. I recommend Reject.

---

> > > > > ### Author Response · Authors · 2023-08-21
> > > > >
> > > > > We acknowledge your emphasis on the need for experimental comparisons. We would like to clarify:
> > > > >
> > > > > - If this refers to comparisons with state-of-the-art text generation methods, in our first response to "Performance Comparison", we have provided experimental comparisons in Figure(d) of the attached PDF, including a comparison with concurrent work GlyphDraw. Note that since there is no publicly available open-source code, checkpoints or APIs for throughout comparison, we try our best to compare with the cases demonstrated in their papers.
> > > > > - If the concern relates to the adaptability to complex characters, we believe it's unreasonable to request additional experiments within the limited rebuttal time. It takes at least one month to collect the Chinese version of the MARIO dataset and retrain our model, while the rebuttal period is only about ten days. Therefore, we have proposed comprehensive potential solutions to adapt the proposed TextDiffuser to other languages with complex characters in our last "Detailed Explanation of the Question" response.
> > > > >
> > > > > If there are any further specific concern, we would highly appreciate detailed clarification and expanation.

---

### Official Review · Reviewer_V9VR · 2023-07-04

**Soundness:** 2 fair
**Presentation:** 4 excellent
**Contribution:** 2 fair
**Rating:** 7
**Confidence:** 4

**Summary:**

The paper proposes TextDiffusers, a 2-stage model to generate images with text per the input text prompt. TextDiffusers also supports text inpainting.  The two stages consist of (1) estimation of layout of keywords (inspired by Layout Transformer) to get character-level segmentation masks and (2) image generation conditioned on the character-level segmentation masks.

The paper also contributes to MARIO-10M, a large data set of image-text pairs along with rich OCR annotations. Benchmark MARIO-Eval with 5414 prompts is created from MARIO-10M and few other existing datasets, in order to evaluate quality of rendering text in the image.

Stage 1 consists of novelty such as encoding of width of keywords. Encoding width of keywords improves IoU, especially for shallower Layout Transformer. Stage 2 introduces character-aware loss and extends the denoising loss to include character level segmentation masks.

The paper contains implementation details for reproducibility, comparison against the states of the art demonstrating large improvements in OCR quality metrics, comprehensive ablations studies, several qualitative samples and limitations.


**Strengths:**

S1) Novel approach to generate images with text

S2) New data set for MARIO-10M along with benchmark MARIO-Eval

S3) Comprehensive experiments demonstrating large improvements against various states of the art on OCR metrics


**Weaknesses:**

W1) While the methodology introduced by the paper seems solid, the main weakness is comparison against strong baseline which was also trained on training split from MARIO-10M. Appendix J alludes to equivalency of fine-tuning of pre-trained model, but Stable Diffusion model is not the best when it comes to producing visual text. It is understandable that official code for Imagen, Parti are not available. Perhaps DeepFloyd could have been fine-tuned on MARIO-10M. From Table 4, DeepFloyd is also better on Fidelity (lower FID score). It is not clear if it would perform better than TextDiffusers with this additional fine-tuning.

W2) Information and comparison of model size and training/inference latency are missing.

W3) Appendix N / Figure 14 in the supplementary material includes a handful of sample generations without text. However, it is not clear if TextDiffusers loses its capability to generate such images in the general sense. Qualitative and Quantitative evaluations to test this are missing.


**Questions:**

Please see list of weakness.

Q1) According to Line 188, the maximum length of tokens (L) is limited to 77. Do all captions in the dataset contain <= 77 tokens?

Q2) Related to Q1, DeepFloyd uses T5-XXL which is much more powerful (as observed by Imagen). Although L<=77, do you have insights on the richness of embedding that may give it a higher edge over TextDiffusers?


**Limitations:**

Yes

---

> ### Author Rebuttal · Authors · 2023-08-09
>
> Thank you for the comprehensive review and feedback on our work.
>
> **Comparison with Strong Baselines Fine-tuning with MARIO-10M:**
> As mentioned in Appendix K, DeepFloyd is better on Fidelity due to its use of two super-resolution modules to generate higher resolution images (1024×1024) versus our models' (512×512) and a stronger text encoder (T5 versus our CLIP). While the lack of DeepFloyd's publicly available training code and implementation details restricted us from retraining or reimplementing it, we acknowledge the potential benefits of fine-tuning it using MARIO-10M to improve its visual text generation performance. However, we assert that merely changing the dataset cannot improve DeepFloyd over TextDiffuser. We emphasize the importantce of our unique proposal of the explicit supervision for visual text rendering through character-level segmentation masks. As evidenced in our comprehensive ablation studies, this feature significantly improves text rendering accuracy and is missing in DeepFloyd.
>
> **Model Size and Latency Metrics:**
> Thank you for highlighting this aspect. Our TextDiffuser builds upon Stable Diffusion 1.5 (859M parameters), adding a Layout Transformer in the first stage (+25M parameters) and modifying the second stage (+0.75M parameters), augmenting it by only about 3% in terms of parameters. It trains within about four days on eight Tesla V100 GPUs, resulting in 6.6 seconds per iteration, with an inference latency of 8.5 seconds per image.
>
> **Capability to Generate General Images:**
> We provide qualitative and quantitative evaluations to demonstrate TextDiffuser's generality in generating non-text general images. We compare TextDiffuser with our baseline Stable Diffusion 1.5 as they have the same backbone. For a quantitative evaluation, the FID scores of 5,000 images generated by prompts randomly sampled from MSCOCO are as follows:
>
> | Sampling Steps |  Stable Diffusion  | TextDiffuser |
> |--------|-----------|-------|
> | 50     | 26.47     | 27.72 |
> | 100    | 27.02     | 27.04 |
>
> For a qualitative evaluation, please refer to Figure(c) in the attached PDF, where we show more comparisons. We can see from the table and figure that TextDiffuser is highly competitive. These evaluations demonstrate that TextDiffuser maintains its capability in the general domain, primarily because our training data encompasses large-scale images from diverse real-world scenes.
>
>
> **Token Length in Captions:**
> A significant majority—99% of the captions in our dataset—are are within the 77 tokens limit.
>
> **Richness of DeepFloyd's T5-XXL Embedding:**
> We agree that the T5-XXL encoder, as leveraged by DeepFloyd, holds the potential for richer textual embeddings as we have mentioned in Appendix K. However, as we analyzed in the answer of **Comparison with Strong Baselines Fine-tuning with MARIO-10M**, the absent of explicit visual text layout supervision in DeepFloyd, remains a deciding factor in the performance of visual text rendering.
>
> In conclusion, we're grateful for your recognition of our approach's novelty, the dataset contribution, and the rigor of our experiments. We will revise our paper to clearly emphasize these in light of your instructive comments.

---

> > ### Comment · Reviewer_V9VR · 2023-08-17
> >
> > Thanks to authors for answering my questions and clarifying them. I don't have further questions. I have also reviewed all other reviews and satisfactory response from the authors.

---

### Official Review · Reviewer_sG87 · 2023-07-05

**Soundness:** 4 excellent
**Presentation:** 3 good
**Contribution:** 3 good
**Rating:** 7
**Confidence:** 4

**Summary:**

This work presents an approach to enhance the text rendering ability of a text-to-image diffusion model. The authors introduce a new diffusion model called text-diffuser that leverages image captions and text segmentation masks to generate text images. They also collect a large-scale image dataset MARIO-10M, which includes explicit text information such as segmentation masks and OCR annotations. The experimental results indicate a significant improvement in text generation ability but a slight decrease in image quality.

**Strengths:**

* The large-scale text image dataset with text annotations MARIO-10M is commendable is valuable for further research in this domain.
* Incorporating character-level segmentation masks to enhance the text rendering ability of a diffusion model is a novel idea. The results presented in the paper demonstrate the effectiveness.

**Weaknesses:**

* The paper mentions that the layout generation model is trained to generate text segmentation masks, which might not be suitable for scene texts with significant perspective changes.  Most of the generation examples lack realistic text styles with perspective changes and complicated text layouts.
* Some GAN-based works [1, 2, 3] exploring scene text editing and a recent diffusion-based scene text editing work [4] could be relevant for part-image generation evaluation. Authors could consider incorporating a discussion and comparison with these related works.


[1] Wu, et al . “Editing Text in the Wild.” https://doi.org/10.1145/3343031.3350929.

[2] Qu, et al. “Exploring Stroke-Level Modiﬁcations for Scene Text Editing,” https://ojs.aaai.org/index.php/AAAI/article/view/25305

[3] Krishnan, et al. “TextStyleBrush: Transfer of Text Aesthetics from a Single Example.”  http://arxiv.org/abs/2106.08385.

[4] Ji, et al. “Improving Diffusion Models for Scene Text Editing with Dual Encoders.” http://arxiv.org/abs/2304.05568.



**Questions:**

* It would be interesting if we can control the style of generated texts through language descriptions on textdiffuser. For example, the image s generated with this caption: a bear holding a board with a **red/purple/yellow** "hello world"? It would be beneficial if the authors could provide some generation examples showcasing this capability.

**Limitations:**

Authors discussed the limitation of the trained model, including small text generation and multiple-word generation.

---

> ### Author Rebuttal · Authors · 2023-08-09
>
> Thank you for your high recognition of our work, particularly the novelty of our TextDiffuser and the significant contribution of the MARIO-10M dataset. We've responded to your comments as follows:
>
> **Layout generation for scene text with perspective changes:**
> Our MARIO-10M dataset reveals that about 90% of the text regions maintain a horizontal orientation with rotation angles smaller than 5 degrees without perspective changes. Hence, our layout generation model is designed to predict horizontal bounding boxes by detecting the coordinates of their left-top and bottom-right points. Adapting our model to predict more realistic scene text is indeed feasible by detecting enhanced coordinates, such as eight coordinates for four points. We'll clarify this adaptability in our revised paper.
>
> **Discussion and comparison of text-editing and part-image generation:**
> We appreciate your suggestion of relevant works [1,2,3,4] in the scene text editing domain. We have taken [1] as a representative work to discuss the differences between text editing and text inpainting (part-image generation), as detailed in Footnote 1 and Appendix O. While we have cited [1,2], we will also incorporate references [3,4] in our updated version.
>
> **Control over the style of generated texts through language descriptions:**
> We are happy to showcase TextDiffuser's innovative capability in controling style of generated texts through language descriptions. As shown in Figure(b) in the attached PDF, TextDiffuser's successfully generates texts in varying colors aligned with the language descriptors.
>
> We will incorporate these insightful discussions in the final version to inspire future explorations and applications in the field.

---

> > ### Comment · Reviewer_sG87 · 2023-08-16
> >
> > Thanks for the response. All my concerns are addressed. It would be beneficial if authors could further improve layout generator in future work.

---

### Official Review · Reviewer_cdWT · 2023-07-07

**Soundness:** 3 good
**Presentation:** 3 good
**Contribution:** 3 good
**Rating:** 6
**Confidence:** 5

**Summary:**

This paper introduces a model for generating visually appealing and coherent text within diffusion models. It also presents the MARIO-10M dataset and the MARIO-Eval benchmark for evaluating text rendering quality. Experimental results demonstrate their method's flexibility and controllability in creating high-quality text images and performing text inpainting.

**Strengths:**

The strengths of this paper include the proposal of TextDiffuser, a flexible and controllable framework based on diffusion models, with two stages for layout generation and fine-tuning.

**Weaknesses:**

The paper does not explicitly mention how the model handles the generation of rich-text images when the number of queries is limited. However, it raises concerns about the efficiency of the model when dealing with a large number of queries.

When there is a sequential relationship between boxes in the context of generating rich-text images, the paper does not specifically address how the model handles the ordering of boxes.

While part-image generation seems reasonable, it could be computationally slow when dealing with a large number of texts as it requires predicting numerous positions.

The paper does not mention the specific number of texts per individual image in the mentioned databases in the main paper, which could be important information to provide.

Regarding image resolution, the paper suggests that it can be low when dealing with single texts. However, when generating a large number of texts, high-resolution generation becomes necessary. It is essential for the authors to address how they ensure the accuracy of each generated text and the efficiency of text generation in such cases.

Using sequential generation for layout masks is an interesting point. Have the authors considered directly using diffusion to predict layout masks? It would be beneficial to have more comparative analysis between the two methods.

**Questions:**

See the weakness

**Limitations:**

The rationale behind this paper, using seq-2-seq generation, is reasonable. However, there is not a clear explanation of how the generation of rich-text is achieved.

---

> ### Author Rebuttal · Authors · 2023-08-09
>
> Thank you for your review and feedback. We've taken care to address each of your concerns below.
>
> **Handling of rich-text images with limited queries & efficiency with a large number of queries:**
> Our first stage of Layout Generation leverages an auto-regressive Transformer whose prediction time correlates with the number of queries (keywords). Meanwhile, the second stage of image generation is independent of the number of queries. The time cost of these two stages, based on experiments conducted three times, are shown below:
>
> | #keywords | Layout Generation (s)|  Image Generation (s) |
> |-----------|--------------|--------------|
> | 1         | 1.07±0.03    | 7.12±0.77    |
> | 2         | 1.12±0.09    | 7.12±0.77    |
> | 4         | 1.23±0.13    | 7.12±0.77    |
> | 8         | 1.57±0.12    | 7.12±0.77    |
> | 16        | 1.83±0.12    | 7.12±0.77    |
> | 32        | 1.95±0.28    | 7.12±0.77    |
> | ...       | ...          | 7.12±0.77    |
>
> As illustrated in the table, our model's efficiency when handling a large number of queries is commendable, primarily due to our use of a two-layer Layout Transformer. What's more, it's desirable to provide a layout template for image generation, which incurs no time cost in model layout prediction.
>
> **Handling ordering of boxes in rich-text images:**
> Our Layout Generation process considers text keywords in the order provided by the user and outputs their box positions accordingly. While the box generation is sequential, the resulting image box order is spatial (two-dimensional). This spatial box ordering is learnt by large-scale real-world images (e.g., MARIO-10M) through our Layout Transformer in the context of rich-text images.
>
> **Computational efficiency in part-image generation with abundant text:**
> To clarify, our part-image generation occurs during the second stage of image generation, which is guided by a predefined text position mask. Thus, there's no added time cost associated with position prediction. For efficiency regarding a large number of queries in the first Layout Generation stage, plese see our first response point above.
>
> **Specific number of texts per image in databases:**
> We appreciate your suggestion. In our revised paper, we will integrate the following table detailing the word count per image in our proposed MARIO-10M dataset:
>
> | #Words |  #Images  | Ratio |
> |--------|-----------|-------|
> | 1      | 592,153   | 5.9%  |
> | 2      | 1,148,481 | 11.5% |
> | 3      | 1,508,185 | 15.1% |
> | 4      | 1,610,056 | 16.1% |
> | 5      | 1,549,852 | 15.5% |
> | 6      | 1,430,750 | 14.3% |
> | 7      | 1,229,714 | 12.3% |
> | 8      |   930,809 |  9.3% |
>
> We can see from the table that most images contain between 3 to 6 words, while images with one word or eight words are less frequent.
>
>
> **The Accuracy and Efficiency of High-resolution Image Generation:**
> We agree that high-resolution generation enhances the quality of rich-text and small-text image generation. Our TextDiffuser can seamlessly transition to high-resolution image generation by deploying a more advanced diffusion backbone. For instance, when we augment the image resolution from 512x512 to 768x768 using Stable Diffusion 2.1 (instead of 1.5), the latent space resolution also increases from 64x64 to 96x96, enhancing our character-level representation. Please refer to Figure(a) in the attached PDF for a visual demonstration of the *accuracy* of our generated text. The inference latency for 512x512 and 768x768 resolutions are 8.5s and 12.0s respectively with a batch size of 1. Thanks to our employment of an *efficient latent* diffusion model, the time increment is insignificant.
>
> **Direct diffusion prediction for layout masks versus sequential generation:**
> In our initial investigations, we explored diffusion models for layout generation. However, we encountered overlapping problems during mask generation for *consecutive characters* and *multiple lines of text*, as illustrated in "Figure(e) in the attached PDF". This overlap stems from the non-autoregressive nature of diffusion models, which generate all text simultaneously. To address this challenge, we turned to the auto-regressive Transformer for layout generation.
>
> We believe that our contributions, including the TextDiffuser model, the MARIO-10M dataset and the MARIO-Eval benchmark, push the boundaries of current text image generation. We appreciate your constructive feedback and hope these clarifications address your concerns. We will include these discussions in our revised paper.

---

> > ### Comment · Reviewer_cdWT · 2023-08-16
> >
> > I appreciate the authors' efforts in answering my questions. I am generally accepting of the current response, and I'd like to raise my rating.

---

### Author Rebuttal · Authors · 2023-08-09

Thank you for taking the time to review. Enclosed in the attached PDF, we have provided some figures for reference.

* Figure(a): Pre-trained on high-resolution Stable Diffusion 2.1 significantly enhances the legibility of small text.
* Figure(b): Demonstration of using language descriptions to control the style of text.
* Figure(c): Visualizations of general images generated by Stable Diffusion 1.5 and the proposed TextDiffuser.
* Figure(d):  Comparison with Character-Aware Model and the concurrent GlyphDraw. The code is not available
for Character-Aware Model, the checkpoints and datasets are not available for GlyphDraw. So we compare the samples in their papers.
* Figure(e):  Layout generation using diffusion model and Transformer.

---

### Decision · Program_Chairs · 2023-09-21

**Decision:**

Accept (poster)

**Comment:**

This paper presents a new method TextDiffuser to improve diffusion models' visual text generation ability by including the text layout in the diffusion generation process. It presents a large-scale text-rich image dataset MARIO-10M and a benchmark MARIO-Eval to measure text image quality, which could be useful in this field. Compared to other SOTA diffusion models, experiment results demonstrate the superior performance of TextDiffuser in generating correct visual text and high-quality images. All reviewers agree on the effectiveness of the proposed method and the contribution of the proposed text image dataset. During the rebuttal period, reviewers cdWT, sG87, and V9VR were satisfied with the author's response about the design of the text-layout generator (cdWT, sG87), style-text image (sG87), and non-text image generation (V9VR) and recommended acceptance. Reviewer AzyM questioned whether this work can be applied to complex characters such as Chinese characters and still recommended rejection after rebuttal.  However, the AC and other reviewers feel this limitation is beyond the scope of this work. The lack of this experiment does not affect the soundness and contribution of this work.